# Nemacol is a small molecule inhibitor of *C. elegans* vesicular acetylcholine transporter with anthelmintic potential

Sean Harrington[1,2,3,13], Jacob Pyche[1,2,13], Andrew R. Burns [2,3], Tina Spalholz[4], Kaetlyn T. Ryan [5], Rachel J. Baker [6], Justin Ching [6], Lucien Rufener[7], Mark Lautens [6], Daniel Kulke[8,10,11], Alexandre Vernudachi[9], Mostafa Zamanian [5], Winnie Deuther-Conrad [4], Peter Brust [4,12] & Peter J. Roy [1,2,3] ✉

Nematode parasites of humans and livestock pose a significant burden to human health, economic development, and food security. Anthelmintic drug resistance is widespread among parasites of livestock and many nematode parasites of humans lack effective treatments. Here, we present a nitrophenyl-piperazine scaffold that induces motor defects rapidly in the model nematode *Caenorhabditis elegans*. We call this scaffold Nemacol and show that it inhibits the vesicular acetylcholine transporter (VAChT), a target recognized by commercial animal and crop health groups as a viable anthelmintic target. We demonstrate that it is possible to create Nemacol analogs that maintain potent in vivo activity whilst lowering their affinity to the mammalian VAChT 10-fold. We also show that Nemacol enhances the ability of the anthelmintic Ivermectin to paralyze *C. elegans* and the ruminant nematode parasite *Haemonchus contortus*. Hence, Nemacol represents a promising new anthelmintic scaffold that acts through a validated anthelmintic target.

Nematodes that parasitize humans and non-human animals including livestock are a significant burden to human health, food security and economic development. Unfortunately, most frontline anthelmintic molecules suffer from inadequacies. For example, anthelmintics used to treat human hookworm (*Ancylostoma duodenale* and *Necator americanus*) and whipworm (*Trichuris trichiura*) are recognized to have inadequate efficacy[1]. Furthermore, the top four marketed anthelmintics used in non-human animals, which include macrocyclic lactones (e.g. Ivermectin), tetrahydropyrimidines (e.g. Pyrantel), imidazothiazoles (e.g. Levamisole) and benzimidazoles (e.g. Thiabendazole), have demonstrated pervasive resistance in cattle and small-ruminants[2] as well as companion animals[3–7]. The dire need for new anthelmintics has been recognized by academic, industry and governmental experts for some time[1,8,9]. Towards identifying novel candidate anthelmintic scaffolds, our group has recently carried out small molecule screens for those that affect the motor activity of the free-

[1]Department of Pharmacology and Toxicology, University of Toronto, Toronto, ON M5S 1A8, Canada. [2]The Donnelly Centre for Cellular and Biomolecular Research, University of Toronto, Toronto, ON M5S 3E1, Canada. [3]Department of Molecular Genetics, University of Toronto, Toronto, ON M5S 1A8, Canada. [4]Department of Neuroradiopharmaceuticals, Institute of Radiopharmaceutical Cancer Research, Helmholtz-Zentrum Dresden-Rossendorf, 04318 Leipzig, Germany. [5]Department of Pathobiological Sciences, University of Wisconsin-Madison, Madison, WI, USA. [6]The Department of Chemistry, University of Toronto, 80 St. George Street, Toronto, ON M5S 3H6, Canada. [7]INVENesis Sàrl, Route de Neuchâtel 15A, 2072 St Blaise (NE), Switzerland. [8]Research Parasiticides, Bayer Animal Health GmbH, Monheim, Germany. [9]INVENesis France Sàrl, 147 Avenue André Maginot, Tours 37100, France. [10]Present address: Department of Biomedical Sciences, Iowa State University, Ames, IA 50011, USA. [11]Present address: Global Innovation, Boehringer Ingelheim Vetmedica GmbH, Binger Str. 173, 55218 Ingelheim am Rhein, Germany. [12]Present address: The Lübeck Institute of Experimental Dermatology, University Medical Center Schleswig-Holstein, 23562 Lübeck, Germany. [13]These authors contributed equally: Sean Harrington, Jacob Pyche. ✉e-mail: peter.roy@utoronto.ca

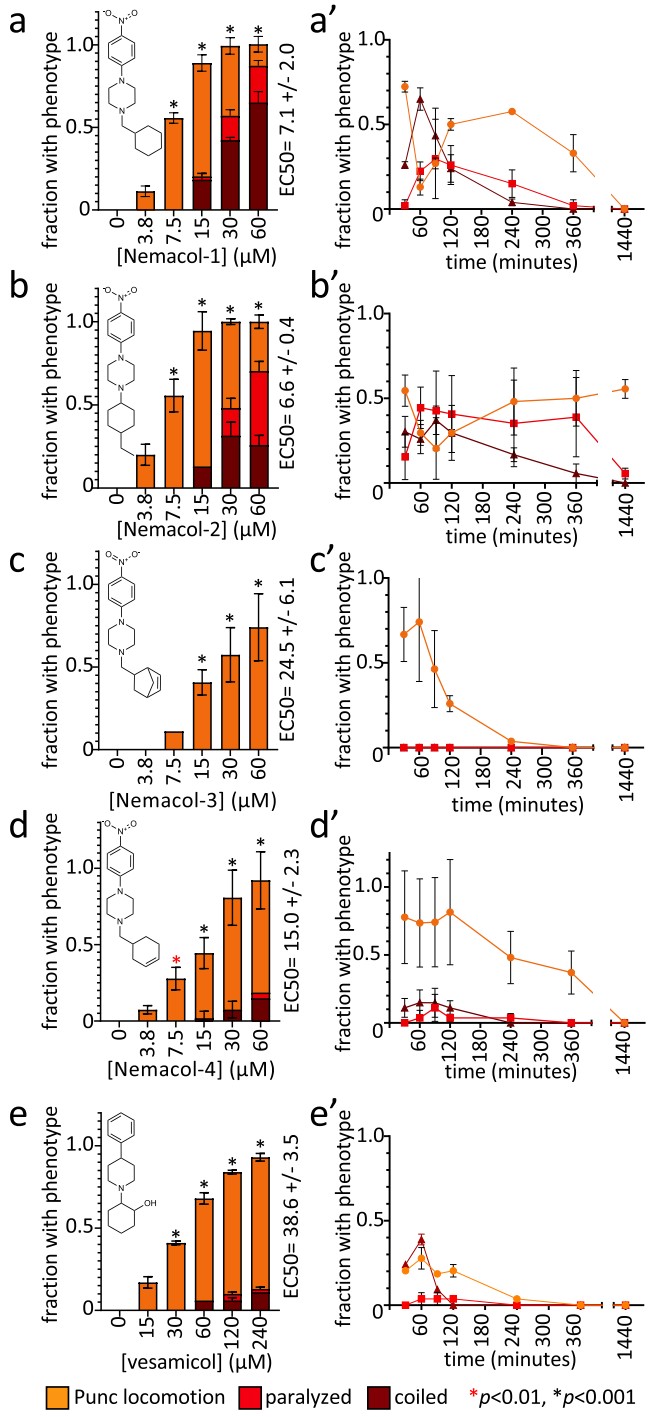

**Fig. 1 | Nemacol analogs induce a shared set of phenotypes with Vesamicol.** Young adult wild-type worms were picked onto solid agar containing the indicated concentration at 60 min (left panels) and scored at the indicated time on 60 μM of indicated compound (right panels). On the right of the left-most graphs, the EC50 and symmetrical 95% confidence interval is shown. Data shown for the 24-h time point were generated independently along with corresponding no drug control. The indicated phenotypes are scored based on subjective classification (see 'Methods' for details). Data are the mean of three biological replicates ($N = 3$) scoring -18 animals ($n = 18$) per trial showing SEM. Significance was calculated using Fisher's method combining one-sided Fisher Exact Test values from each biological replicate. A red asterisk indicates $P < 0.01$ and a black asterisk indicates $P < 0.001$. The $P$ values in (**a**) relative to 0 μM Nemacol-1 are 5.4E-9 (7.5 μM), 2.1E-22 (15 μM), 2.5E-28 (30 μM), 2.5E-28 (60 μM). The $P$ values in (**b**) are 7.3E-28 (7.5 μM), 3.6E-8 (15 μM), 6.9E-23 (30 μM), 2.5E-26 (60 μM). The $P$ values in (**c**) are 8.2E-6 (15 μM), 9.4E-10 (30 μM), 4.7E-14 (60 μM). The $P$ values in (**d**) are 3.0E-3 (7.5 μM), 1.1E-5 (15 μM), 9.8E-17 (30 μM), 1.4E-22 (60 μM). The $P$ values in (**e**) are 2.3E-5 (7.5 μM), 1.4E-11 (15 μM), 1.0E-17 (30 μM), 4.5E-22 (60 μM).

carbamates that in turn lead to catastrophically high levels of synaptic ACh. Finally, vesicular acetylcholine transporter (VAChT), which packages ACh into presynaptic vesicles[12,25,26], is inhibited by Vesamicol and by the relatively novel spiroindoline scaffold[27]. Spiroindolines have been investigated as nematode and/or insect parasiticides by Zoetis (patent US20160296499A1[28]), Intervet Inc (patent US9096599B2[29]) and Syngenta (patent US9174987B2[30]), but none have yet brought a spiroindoline to market[27].

Here, we present a novel nitrophenyl-piperazine scaffold that inhibits VAChT. Because of its structural similarity to the VAChT inhibitor Vesamicol (which is used as a tool compound[25,27]), we call this scaffold Nemacol. Nemacol rapidly induces *C. elegans* worms to coil their bodies, which is a phenotype shared by mutants of the CHA-1 choline acetyltransferase enzyme, which makes ACh, and mutants of the VAChT worm ortholog (UNC-17)[12,26]. Herein, we describe the kinetics of the Nemacol-induced phenotypes and provide chemical-genetic and biochemical evidence to show that Nemacol inhibits nematode VAChT. We demonstrate that Nemacol also disrupts the motor activity of the commercially important animal parasites *Dirofilaria immitis* (dog heartworm) and *Haemonchus contortus* (a nematode parasite of ruminants). We also demonstrate that Nemacol can enhance the macrocyclic lactone Ivermectin to kill *C. elegans* and *H. contortus*. Finally, we show that select Nemacol analogs can maintain their low micromolar potency in nematodes but reduce their affinity for the mammalian VAChT receptor tenfold relative to the Nemacol parent, demonstrating the potential for an expanded therapeutic window. We conclude that Nemacol is a novel small molecule scaffold with anthelmintic potential.

## Results

### Nemacol inhibits the vesicular acetylcholine transporter

From our previous screen[10], we identified four structurally similar molecules that stimulate *C. elegans* egg-laying (Egl) and elicit similar uncoordinated motor phenotypes (Fig. 1). All four molecules, which were referred to as wact-45, wact-6, wact-46, and wact-47, share a 1-ethyl-4-(4-nitrophenyl)piperazine substructure. We have renamed these molecules Nemacol-1, −2, −3, and −4, respectively. Within minutes of exposure, Nemacol-1 causes frequent abrupt pausing during locomotion, a phenotype that we will refer to here as 'pausing <u>unco</u>ordinated (Punc) locomotion' (Fig. 1a–d and Supplementary Movies 1–3). Over the course of 4 h, the Punc phenotype transits to tight coiling (Supplementary Movies 4 and 5) and paralysis whereby a fraction of the animals fails to locomote during the observation period even when prodded on the head (see Supplementary Movies 6 and 7) and then gradually transits back to the Punc locomotion. After 24 h of being on Nemacol-1, animals no longer have obvious motor defects (Fig. 1a'). Blocking drug-metabolizing cytochrome P450s by knocking

living nematode *Caenorhabditis elegans*[10]. One scaffold that we focus on here inhibits cholinergic signalling. The neurotransmitter acetylcholine (ACh) is the primary signalling molecule in most animals that triggers muscle contraction[11]. Without careful regulation of ACh, animals lose muscular control and die[12–14]. The essentiality of cholinergic signaling is a major point of vulnerability that has been repeatedly exploited both in nature as a target of toxins and venoms[14,15] and by humans as a pesticide strategy[16,17].

Three protein groups within the cholinergic pathway are targeted by anthelmintic and nematicidal molecules. These include nicotinic ACh receptors (nAChRs), which are agonized by the imidazothiazoles and tetrahydropyrimidines[3,18–21] and antagonized by Derquantel[22–24]. Acetylcholinesterases (AChE), which break down ACh at the neuromuscular junction and are inhibited by organophosphates and

**Legend (Fig. 1):** Punc locomotion | paralyzed | coiled | *p<0.01, *p<0.001

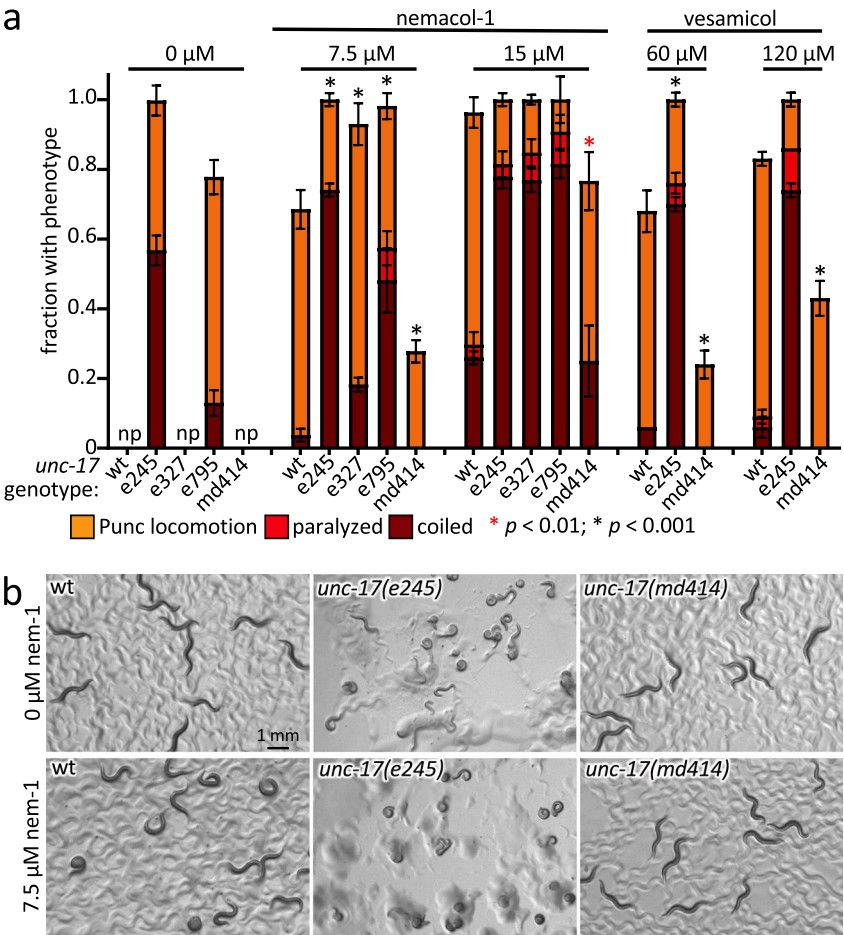

**Fig. 2 | Nemacol induces phenotypes consistent with the inhibition of UNC-17 (VAChT). a** Adult wild-type (wt) worms or *unc-17* mutants with the indicated alleles (*e245*, *e327*, *e795*, and *md414*) were picked onto solid agar containing either Nemacol-1 or Vesamicol at the indicated concentration at 60 min. The legend details are the same as that for Fig. 1 except that $N = 3$ and $n = 18$. Significance was calculated using Fisher's method (one-sided comparisons; see 'Methods'). Np; the animals fail to exhibit any of the Punc, paralyzed or coiled phenotypes. A red asterisk indicates $P < 0.01$ and a black asterisk indicates $P = 0.001$. The *P* values for comparison to the wt control at the same drug concentration are as follows: At 7.5 μM Nemacol-1, 1.9E-4 (*e245*), 3.7E-4 (*md414*), 6.0E-6 (*e327*), and 9.7E-17 (*e795*); at 15 μM Nemacol-1, 7.0E-3 (*md414*); at 60 μM vesamicol, 2.0E-4 (*e245*), 6.0E-5 (*md414*); and at 120 μM vesamicol, 5.3E-5 (*md414*). **b** Photomicrographs of worms of the indicated genotype. The drug condition for the top and bottom rows is indicated on the left. Worms were incubated on the plates for 60 min. The scale in the upper left is identical for all panels. Images are a representation of observations made in (**a**) that were conducted over $N = 3$ biological replicates.

down *C. elegans* cytochrome P450 reductase (EMB-8) antagonizes both the metabolism of Nemacol-1 ($P < 0.05$) and the dissipation of these phenotypes ($P < 0.001$) (Supplementary Fig. 1). Effectiveness of the different analogs likely reflects their different rates of accumulation, metabolism, detoxification and target engagement.

Of the four original Nemacol hits, analogs 1 and 2 induced the strongest phenotypes in *C. elegans*. However, Nemacol-2 demonstrated adverse effects in our previous counter-screens[10]. We therefore focused on Nemacol-1 for much of the analyses presented below.

The phenotypes induced by the Nemacol scaffold are shared with *C. elegans* mutants that have defective cholinergic signaling. For example, mutations in the CHA-1 choline O-acetyltransferase that produces ACh or in the UNC-17 VAChT that packages ACh into presynaptic vesicles exhibit a characteristic coiling phenotype[12,27]. Furthermore, the Nemacol scaffold has structural similarity to the canonical VAChT inhibitor Vesamicol, which induces coiling and Punc locomotion phenotypes, but with an EC50 that is 5.4-fold higher than Nemacol-1 ($P = 2.6E-6$) (Fig. 1). These similarities led to the hypothesis that Nemacol inhibits *C. elegans* UNC-17/VAChT. We tested this hypothesis in four ways.

First, we reasoned that if UNC-17 is Nemacol's target, then weak alleles of *unc-17* (*e327* and *e795*)[26,31] should be hypersensitive to the compound, which is what we observed ($P < 0.001$) (Fig. 2a).

Second, the *C. elegans* UNC-17 C391Y mutation called *md414* had previously been shown to disrupt Vesamicol binding[31,32]. We reasoned that if Nemacol inhibits VAChT through a shared binding site with Vesamicol, then *unc-17(md414)* mutant animals should suppress Nemacol phenotypes, which is what we observed ($P < 0.001$) (Fig. 2a, b).

Third, we tested whether Nemacol can suppress the paralysis induced by inhibitors of AChE. AChE inhibition results in excess ACh at neuromuscular junctions, which in turn paralyzes the worm due to excess muscle contraction[32,33]. Reduction-of-function *unc-17* mutants are known to resist the effects of AChE inhibitors[26,27,32]. If Nemacol inhibits UNC-17 and results in lower levels of ACh at the synaptic cleft, then Nemacol treated worms should also resist the effects of AChE inhibitors. Indeed, Nemacol-1 can suppress the paralysis induced by two structurally distinct AChE inhibitors, namely trichlorfon (an organophosphate) and aldicarb (a carbamate) in a dose-dependent manner (Fig. 3 and Supplementary Fig. 2). We find that 30 μM Nemacol-1 yields nearly an identical 2.2-fold shift in the EC50 of trichlorfon paralysis compared to 250 μM Vesamicol representing an 8.3-fold shift in potency ($P < 1E-15$) (Fig. 3c). The ability of Nemacol-1 to prevent the paralysis induced by AChE inhibitors is obvious by simply looking at the animals (Fig. 3d).

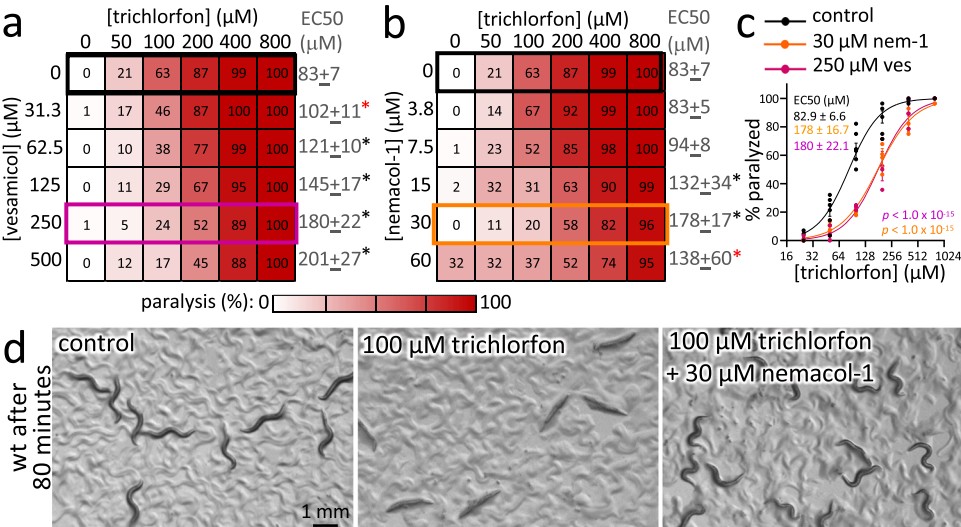

**Fig. 3 | Nemacol suppresses the paralysis induced by the acetylcholinesterase inhibitor Trichlorfon. a, b** Double dose-response matrices of Vesamicol + trichlorfon (**a**) and Nemacol-1 + trichlorfon (**b**) showing the fraction of animals that were scored as paralyzed after 80 min. Values in cells represent the % of animals scored as paralyzed. Data are the mean of 3 biological replicates ($N = 3$; $N = 6$ for the 0 μM nemacol/vesamicol dose-responses of trichlorfon) with $n = 28$ animals per condition. The EC50s with the symmetrical 95% confidence interval for the trichlorfon dose response in the background of indicated Vesamicol or Nemacol concentration is shown to the right of the respective rows. Significant differences of the EC50s relative to the trichlorfon-only dose response are calculated using extra sum-of-squares F test and are indicated with a red ($P < 0.01$) or black ($P < 0.001$) asterisks. The P values of the trichlorfon dose-response with increasing concentrations of vesamicol (relative to the trichlorfon dose-response without vesamicol) are 3.3E-3 (31.3 μM vesamicol), 1.1E-7 (62.5 μM vesamicol), 7.7E-11 (125 μM vesamicol), 4.0E-15 (250 μM vesamicol), and <E-15 (500 μM vesamicol). The P values

of the trichlorfon dose-response with increasing concentrations of Nemacol-1 (relative to the trichlorfon dose-response without Nemacol) are 5.9E-5 (15 μM Nemacol), <E-15 (30 μM Nemacol), 2.3E-3 (60 μM Nemacol). **c** Dose-response curves from the black, fuchsia, and orange boxes in (**c**, **d**) that highlight the shift in the EC50 of trichlorfon paralysis +/− 250 μM Vesamicol (fuchsia) or 30 μM Nemacol-1 (orange) relative to the trichlorfon-only control (black) reporting the SEM. The P values of the trichlorfon dose response with 250 μM Vesamicol relative to without ($P < 1$E-15) and the trichlorfon dose response with 30 μM Nemacol-1 relative to without ($P < 1$E-15) are shown. The EC50s with the 95% confidence interval are shown in the inset. Values in cells represent the % of animals scored as paralyzed. Data are the mean of three biological replicates ($N = 3$; $N = 6$ for the 0 μM nemacol/ vesamicol dose-responses of trichlorfon) with $n = 28$ animals per condition. P values are calculated using an extra sum-of-squares F test. (**d**) Details on the photomicrographs are identical to that described for (**b**) above.

Finally, we tested whether Nemacol was able to directly interact with vertebrate VAChT. We measured the ability of Nemacol-1 to displace [³H]Vesamicol bound to rat VAChT expressed in rat PC12[A123.7] cells, which is an assay previously established for measuring small molecule affinity for VAChT[31,34]. We found that Nemacol-1 binds mammalian VAChT, but with 37-fold less affinity than Vesamicol ($P < 1$E-15) (Fig. 4a). Together, these data indicate that Nemacol likely elicits motor phenotypes in *C. elegans* through the inhibition of UNC-17/VAChT at the Vesamicol binding site.

**Nemacol analogs demonstrate potential selectivity against the nematode VAChT target**

Above, we show that Nemacol-1 is 5.4-fold more potent than Vesamicol in live *C. elegans* worms, but has 37-fold less affinity than Vesamicol for the mammalian VAChT target. This comparison raised the possibility that the Nemacol scaffold could be modified to create nematode-selective VAChT inhibitors. Although co-crystal structures of Vesamicol with VAChT are not available, several residues known to be important for interaction are known[27,31,35,36]. Inspection of a VAChT multiple sequence alignment shows that residues immediately flanking Vesamicol-interacting residues are divergent in nematodes (Supplementary Fig. 3), raising the possibility that nematode-selectivity may be achieved.

We explored the activity of the Nemacol scaffold by testing 50 analogs in acute motor tests in culture against *C. elegans*, the free-living nematode *Pristionchus pacificus*, and with select molecules, against the dog heartworm *Dirofilaria immitis*. 44 of these analogs were procured from commercial sources and 6 other analogs were synthesized by us (see 'Methods'). We found many analogs to be active against *C. elegans* and *Pristionchus*, and two analogs (Nemacol-1 and

Nemacol-5) to have low micromolar activity against *Dirofilaria* (Fig. 4c and Supplementary Figs. 4 and 5). We repeated tests of Nemacol-1 activity against *Dirofilaria immitis* in a second lab and found similar results (Supplementary Fig. 6) (see 'Methods'). Of the 16 analogs tested against *D. immitis* that contained a 4-nitrophenyl group, 8 were active (Fig. 4c; EC50 ≤ 22.5 μM). Furthermore, only analogs containing a nitro group in position $R_1$ were active against *D. immitis*. Together, these observations suggest that the nitrophenyl group is important for activity against *D. immitis*. These results show that the Nemacol scaffold has activity beyond *C. elegans*.

Next, we chose 12 diversely structured analogs that had good activity against *C. elegans* and investigated whether any might have weakened affinity for the rat VAChT relative to Vesamicol and Nemacol-1. Ten of the 12 analogs tested had more than tenfold less affinity to the rat VAChT relative to Vesamicol and six had more than 30-fold less affinity (Fig. 4b, c). Comparing Nemacol-1's in vitro mammalian VAChT Ki (0.77 μM) to its *C. elegans* in vivo EC50 activity (7.1 μM) reveals an activity ratio of 0.11. By contrast, Nemacol-63's equivalent ratio is 1.10, which is an improvement of over tenfold in selectivity. The two analogs with the poorest VAChT affinity (Nemacol-62 and 63) share a cyclopentylmethyl group in the $R_2$ position, suggesting that this feature diminishes affinity with mammalian VAChT. These results suggest that it may be possible to modify the Nemacol scaffold to achieve nematode selectivity.

**Nemacol synthetically interacts with ivermectin**

The nAChR antagonist and anthelmintic derquantel synthetically interacts with abamectin, a macrocyclic lactone and Cl⁻ channel agonist[22,23,37–39]. Nemacol inhibits VAChT and consequently depresses cholinergic signaling by decreasing synaptic vesicle ACh

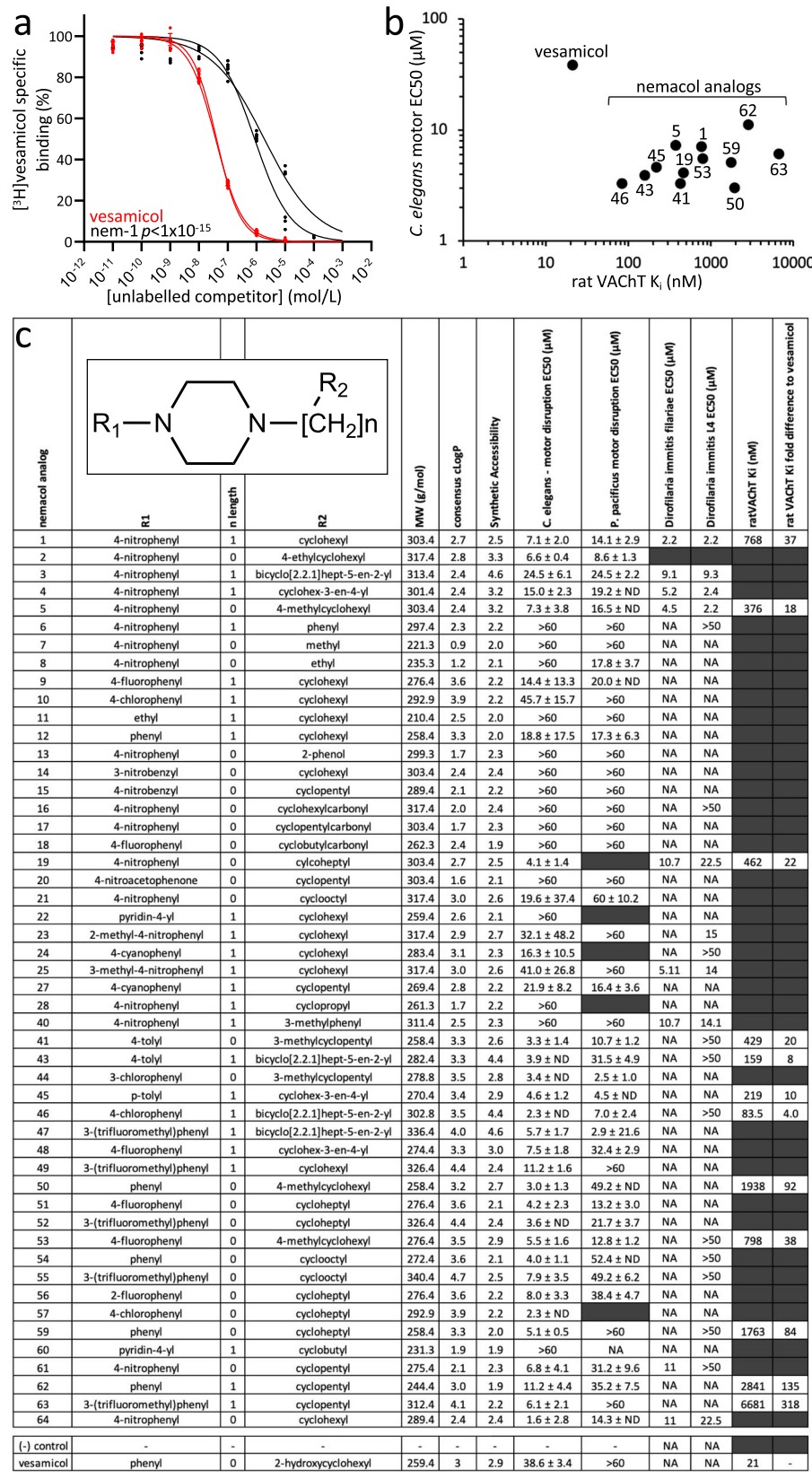

content. We reasoned that Nemacol, like derquantel, might therefore synthetically interact with macrocyclic lactones to disrupt nematode neuromuscular function. Indeed, we found that in 3-day liquid viability assays (see 'Methods'), combinations of Ivermectin and Nemacol-1 could yield effective killing of *C. elegans* at concentrations that had negligible effect on their own (for one example, see 30 μM Nemacol-1 and 15 nM Ivermectin in Fig. 5a). Nemacol-1 significantly lowered Ivermectin's EC50 by 3-fold ($P < 0.001$; Fig. 5a). Furthermore, the combination of the two compounds yields a global Zero Interaction Potency (ZIP) synergy score of 20.0, which exceeds the ZIP score threshold of synergy (10)[40–42] (the intense red area of Fig. 5b).

**Fig. 4 | Nemacol analogs demonstrate nematode-selective activity. a** Nemacol-1 (black) and (-)Vesamicol (red) competitive displacement of [³H] Vesamicol binding of rat VAChT. Shown is the binding curve for each of two biological replicates (*N* = 2) tested in technical triplicate (*n* = 3) reporting the SEM for each binding curve. The *P* value comparison of the two sets of curves was calculated using Fisher's Method ($P < 10^{-15}$ for both; see 'Methods'). The determined $K_i$ and asymmetrical confidence interval for (−)Vesamicol was 19.5 nM (17.7 nM–21.6 nM) and 22.4 nM (17.2 nM–29.4 nM) for each biological replicate. The determined $K_i$ and confidence interval for nemacol-1 was 875 nM (665 nM–5.99 µM) and 661 nM (503 nM–868 nM) for each biological replicate. **b** Comparison of Vesamicol and Nemacol analog in vivo potency and rat VAChT binding affinity. Details of the means and confidence intervals are in (**c**) and Supplementary Table 1. **c** Structure

activity relationship summary of Nemacol analog activity across nematode species and rat VAChT inhibition constants ($K_i$). *C. elegans* and *Pristionchus pacificus* EC50s are derived from biological triplicates (*N* = 3) with technical duplicate (*n* = 2) scoring 18 animals per condition. See Supplementary Figs. 4 and 5 for the dose-response curves. *Dirofilaria immitis* microfilariae immobility EC50 data are from triplicate (*N* = 3) measurement of ~250 animals per condition after 72 h of drug exposure. *D. immitis* L3/L4 immobility EC50 data are derived from singlet observations from 20 freshly isolated L3s after 72 h of incubation (see 'Methods'). NA no activity. The inhibition constant ($K_i$) confidence intervals for interaction with rat VAChT are presented in Supplementary Table 1. The grey cells indicate that the experiment was not done.

Previous work has identified a number of targets for Ivermectin in *C. elegans*, including the glutamate-gated chloride channels AVR-14, AVR-15, and GLC-1[43]. We investigated whether the synergy observed between Nemacol-1 and Ivermectin in wild-type animals is maintained in the *avr-14; avr-15; glc-1* triple *C. elegans* mutant. As expected, the triple mutant lost responsiveness to Ivermectin at the concentrations tested (Supplementary Fig. 7a). For example, Supplementary Fig. 7b compares Ivermectin's effects on the triple mutant (without Nemacol-1; Supplementary Fig. 7a) to Ivermectin's effects on the wild type (without Nemacol-1; Fig. 4a) ($P < 5.5 \times 10^{-11}$). Notably, the double drug treatment remains synergistic in the triple mutant (Supplementary Fig. 7c), albeit 0.6-fold less compared to how the drugs behave in the wild type (compare with Fig. 5b). This suggests that some of the observed synergy between Nemacol-1 and Ivermectin derives from Ivermectin's interaction with other targets beyond AVR-14, AVR-15, and GLC-1. This insight is consistent with previous findings showing that additional components, including GLC-3 and perhaps UNC-7, also mediate Ivermectin sensitivity in *C. elegans*[43–45].

Ivermectin is known to inhibit drug pumps and may block the worm's elimination of Nemacol[46,47]. To test whether the inhibition of VAChT contributed to the synthetic interaction between Nemacol and Ivermectin, we tested whether the *unc-17(e245)* reduction-of-function mutant was more sensitive to the effects of Ivermectin. Indeed, *unc-17(e245)* demonstrated a sensitivity to Ivermectin ($P < 1E-15$ relative to wild type's sensitivity to Ivermectin) that was comparable to 30 µM Nemacol treatment ($P < 1E-15$ relative to wild type's sensitivity to Ivermectin) (Fig. 5c). This suggests that the synergistic interaction is not solely due to altered metabolism or export of Nemacol by Ivermectin.

Finally, we wanted to test whether Nemacol can enhance the effects of Ivermectin in the context of a parasitic nematode. Because *Dirofilaria immitis* is known to be refractory to Ivermectin in in vitro assays[48–50], we asked whether enhancement may be seen with the ruminant nematode parasite *Haemonchus contortus*[51]. A small survey indicated that Nemacol-53 has activity against *H. contortus*, with an EC30 of 26 µM (Fig. 5d). This concentration of Nemacol-53 is able to significantly enhance Ivermectin's ability to paralyze *H. contortus* in vitro ($P = 0.002$) (Fig. 5e). These data indicate that Nemacol may have additional utility in its ability to sensitize nematodes to one of the most widely used anthelmintics in the world.

## Discussion

The current repertoire of anthelmintics available for the control of parasitic nematodes that infect humans and non-human animals is lacking[1,52]. Hence, the identification of novel and selective nematicidal compounds is a key step in protecting human health and food security. Here, we have identified the Nemacol scaffold that disrupts nematode motor function via the inhibition of VAChT. Select Nemacol analogs maintain nematode activity whilst losing affinity for mammalian VAChT, suggesting that nematode selectivity can be achieved with this scaffold. Nemacol is detoxified in *C. elegans* over the course of 24 h. However, assays performed here against multiple parasitic nematodes over an equivalent timespan (or longer) (see Figs. 3c and 4d and

Supplementary Fig. 6) indicate that Nemacol's effects can persist in parasites.

How does Nemacol compare to other VAChT inhibitors in the context of being a candidate anthelmintic lead? Nemacol is similar in structure to Vesamicol, a canonical VAChT inhibitor[53] and both compounds likely interact with a common binding site on VAChT[31]. However, the Vesamicol scaffold has lackluster activity against nematodes, has high affinity for mammalian VAChT, and has low tolerability in rats[27,31,54]. These features likely account for a lack of interest in developing the Vesamicol scaffold as an anthelmintic.

In contrast to Vesamicol, the spiroindoline scaffold has been rigorously pursued as a candidate anthelmintic because it selectively incapacitates nematodes and insects[27,28,55]. Indeed, there has been heavy commercial interest in pursuing the spiroindolines as candidate anthelmintics by several groups, including Zoetis Services LLC[28], Syngenta Ltd[30] and Intervet Inc[29]. The spiroindolines have been shown to inhibit VAChT via residues that are at least partially distinct from those that interact with Vesamicol[27,31]. Of note, Nemacol analogs incapacitate *C. elegans* motor activity at equivalent concentrations to the spiroindolines (compare Figure 6 in ref. [27] to Fig. 4a; note 1 µg/mL SYN351 = 1.9 µM) and Nemacol analogs have also proved comparatively effective at incapacitating *D. immitis* filariae relative to spiroindolines (compare Table 2 in US20160296499A1 to Fig. 4a; note that we report ED50 compared to minimal inhibitory concentration[28]).

The high lipophilicity of the spiroindolines may have so far stifled their development into commercial products. Guidance from the European Union's European Chemical Agency[56] highlights that compounds with a Log P greater than 4 have accumulative potential in adipose tissue of animals. The primary spiroindoline lead pursued by Syngenta (SYN876 in ref. [27]) has a SwissADME predicted consensus LogP of 5.53 suggesting that this lead may have concerning accumulative potential. SwissADME is a highly cited chemoinformatics tool that is widely used in the field[57]. In contrast, 92% of Nemacol analogs had a SwissADME predicted consensus Log $P \leq 4.0$ with a median of 2.93[57] (see Fig. 4c).

Relative to the structural complexity of the spiroindoline scaffold (MW 453.53 with a SwissADME synthetic accessibility score of 4.23)[57] the Nemacol-1 structure is simpler (MW 303.4 with a synthetic accessibility score of 2.52)[57,58] (see Fig. 4c). The higher the synthetic accessibility score, the more difficult the synthesis[57,58]. Indeed, we have found Nemacol to have a relatively inexpensive synthesis route with several analogs so far synthesized requiring only a two-step metal-free synthetic sequence that does not require purification of the intermediate (see 'Methods').

VAChT has clearly been recognized by multiple commercial groups as an attractive anthelmintic target[27–30]. One reason for this is it may be difficult to mutate VAChT to a state that reduces an inhibitor's efficacy without compromising the transporter itself. The VAChT mutant residue that confers resistance to Vesamicol disrupts the transporter's ability to interact with the inhibitor, reduces the transporter's interaction with acetylcholine and lead to motor defects in *C. elegans*[31]. Hence, missense mutations that alter the ability of VAChT

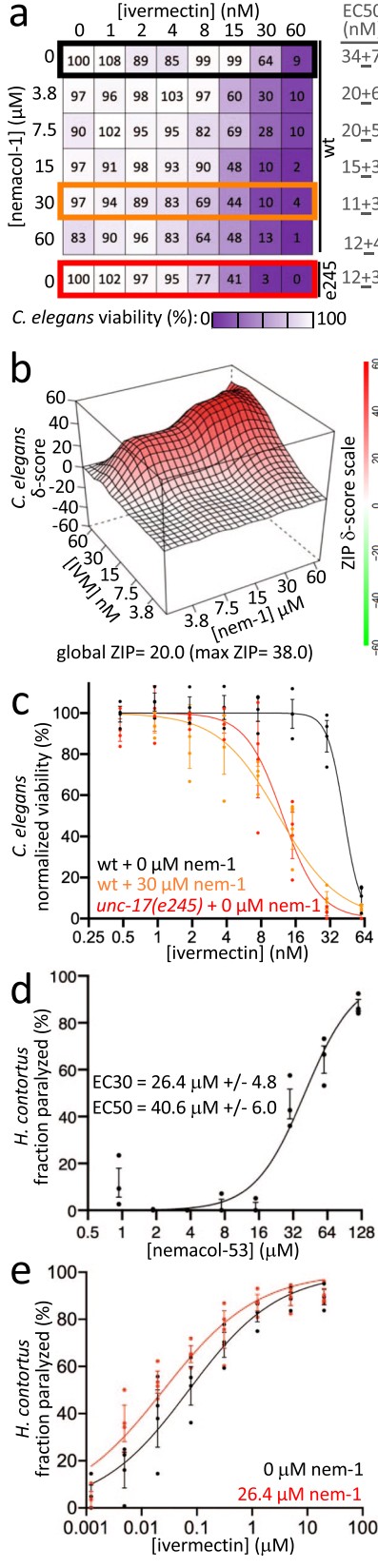

**Fig. 5 | Nemacol-1 synergistically kills *C. elegans* in combination with Ivermectin. a** *C. elegans* viability over a double dose response matrix of Nemacol-1 and Ivermectin reporting the means of the control-normalized fraction of animals alive in treatment wells of 3-day growth assays (see 'Methods'). -20 *C. elegans* L1s were added to wells containing the indicated condition with 0.6% DMSO and an HB101 *E. coli* food source in nematode growth medium. Data are from three biological replicates (*N* = 3) conducted in technical duplicate (*n* = 2). Significant differences of the EC50s relative to the ivermectin-only dose response are calculated using extra sum-of-squares F test. The respective *p* values are as follows: 2.8E-03 (3.8 μM Nemacol), 1.8E-03 (7.5 μM Nemacol), 1.6E-07 (15 μM Nemacol), 7.0E-08 (30 μM Nemacol), 1.4E-06 (60 μM Nemacol), and 7.5E-09 (*e245* without Nemacol). The red and black asterisks indicates *P* < 0.01 and *P* = 0.001, respectively. **b** Zero Interaction Potency (ZIP) synergy score plot of the Ivermectin + Nemacol-1 double-dose response interaction generated using the SynergyFinder2.0 server[42]. **c** Dose-response curves from (**a**) highlighting the EC50 shifts of Ivermectin killing of wild type *C. elegans* in comparisons of no Nemacol treatment (black line, corresponding the values in the black box in (**a**)) versus 30 μM Nemacol-1 (orange line, corresponding the values in the orange box in (**a**)) (*P* = E-15), or no Nemacol treatment (black line) versus *unc-17(e245)* mutants (red line, corresponding the values in the red box in (**a**)) (*P* = 5.0E-15) reporting the SEM. *P* values are calculated using a two-sided extra sum-of-squares F test (see 'Methods'). **d** *H. contortus* control-normalized motility over a dose-response of Nemacol-53 showing the mean of three biological replicates (*N* = 3) and reporting the SEM (see 'Methods'). Four-parameter curves were fit in GraphPad 9.3.1 and used to calculate EC30 and EC50 values with symmetrical 95% confidence intervals. **e** *H. contortus* control-normalized motility over a dose-response of Ivermectin dose-response +/− an EC30 concentration of Nemacol-53 (26.4 μM) showing the mean of three biological replicates (*N* = 3) and reporting the SEM (see 'Methods'). Four-parameter curves were fit in GraphPad 9.3.1 and the associated *P* value (*P* = 0.0019) was calculated using a two-sided extra sum-of-squares F test comparing +/− 26.4 μM Nemacol-53 curves.

Despite the conservation, there are key differences in nematode VAChT sequence relative to non-target species near the presumptive Vesamicol binding site, suggesting that broad-spectrum nematode-selectivity may be possible to achieve with VAChT inhibitors.

In addition to having activity on its own, Nemacol enhances Ivermectin activity in *C. elegans* and the ruminant parasite *H. contortus*. At the present time, the Nemacol scaffold represents only a set of tool compounds. However, its ability to synergize with one of the most effective anthelmintics increases its potential utility in the field and potentially lowers the effective dose needed of Ivermectin and increases its therapeutic window. In addition, Nemacol's ability to synergize with Ivermectin persists even in animals that lack some of Ivermectin's targets, albeit less effectively. These reasons, together with Nemacol's synthetic accessibility and the attractiveness of its target, make Nemacol an important scaffold to further develop as an anthelmintic agent.

## Methods

### Ethics and sex statements

Collaborators working with vertebrate hosts of nematode parasites conducted research complying with relevant ethical regulations. Bayer Animal Health GmbH (Monheim, Germany) operated in accordance with the local Animal Care and Use Committee and governmental authorities (LANUV#200/A176 and #200/A154). The Zamanian group sources their Dirofilarial nematodes from the NIH FR3 (BEI Resources) resource center. Animal research at the FR3 complies with all relevant ethical regulations and operates under the University of Georgia IACUC (AUP#: A2019 04-010-Y1-A0). The INVENesis group performs animal experimentations in the Infectiology of Farm, Model, and Wildlife Animals Facility (PFIE, Centre INRAE Val De Loire, D371753). Experimental protocols were designed in compliance with French law (2010/63/EU, 2010; Rural Code, 2018; Decree No. 2013-118, 2013) concerning the use of laboratory animals. Care and euthanasia of animals were practiced according to the national ethical guidelines and approved by the local ethics committee for animal

inhibitors to interact with the transporter will likely confer a clear selective disadvantage to worms in the absence of Nemacol pressure. Ensuring persistence of drug-sensitive alleles within population is becoming a more common practice of parasite management[59]. A second reason for the keen interest in the VAChT target may be because of its high conservation across nematodes[60] (Supplementary Fig. 3).

experimentation (Comité d'Ethique en Expérimentation Animale Val de Loire, CEA VdL N°19): APAFIS#17560. The authors are committed to the principles of the 3Rs: reduction, refinement, and replacement of experimental animals.

The sex of *C. elegans* used in the experiments is hermaphroditic. There are no visible markers of sex differentiation in the *Dirofilaria* or *Haemonchus* larvae used herein and there is no evidence of drug effects unevenly distributed across larvae. Given the numbers of *Dirofilaria* and *Haemonchus* used in the Kulke, Zamanian and INENesis experiments, both sexes were likely well-represented.

### Worm culture, strains, and photography

All nematode strains were cultured using standard methods at 20 °C unless otherwise indicated[61,62]. The N2 (wild-type) strain of *Caenorhabditis elegans* and *C. elegans* mutant strains were obtained from the *C. elegans* Genetic Center (University of Minnesota). Synchronization of worm developmental stages was achieved using standard bleach preparation protocols[63]. Synchronized young adult animals are acquired by incubating synchronized L1s (first larval stage animals hatched overnight in M9 buffer[62] on plates with *E. coli* (strain OP50) food for 64–72 h at 20 °C. Worms were maintained on Modified Youngren's, Only Bacto-peptone (MYOB) media containing 2% agar with a surface lawn of OP50 strain *Escherichia coli*[64]. Photomicrographs shown in Fig. 2 were taken on a Leica MZ16FA stereodissection microscope with an EC4 camera and processed with Leica LAS EZ and ActivePresenter software (Atomi Systems, Inc).

### Preparation of small-molecules in solid media

Molten Modified Youngren's, Only Bacto-peptone (MYOB) containing 2% agar was equilibrated to 55 °C in a water bath. Small-molecules solvated in DMSO were spiked into at least 4 mL of molten media mixture in 15 mL conical tubes, inverted five times and vortexed. The final concentration of dimethyl sulfoxide (DMSO) in each of the wells was 1% v/v. One mL of media containing small-molecule was pipetted into wells of a 24-well plate using a 10 mL Sarstedt serological pipette. Plates were dried under sterile air flow in a laminar flow hood for 90 min. After 90 min 25 μL of OP50 strain *E. coli* bacteria culture from a saturated Luria Broth culture was added by pipette onto the surface of the culture media. Plates were allowed to dry on a benchtop proximal to flame for 15 min. Plates were covered and wrapped in tinfoil and were used the following day.

### Scoring of motor phenotypes

Locomotor phenotype analyses were done in 24-well plates with 1 mL of MYOB substrate (27.5 g Trizma HCl, 12 g Trizma Base, 230 g bacto tryptone, 10 g NaCl, 0.4 g cholesterol (95%)) seeded with 25 μL of OP50 *Escherichia coli* on each well. Each compound was added to the MYOB substrate before pouring to achieve the desired final concentrations of 30 μM or 60 μM after diffusion through the media. Synchronized young adult worms are transferred into each well using a platinum wire pick. A Leica MZ75 stereomicroscope was used to visualize the movement of worms on the solid substrate. The specific dominant locomotor phenotype (i.e. 'Punc', 'paralyzed', 'coiler' or 'wild-type locomotion') was scored after the touch on the head with a platinum wire over a ~3–5 s of observation. Animals were scored as having either Punc locomotion if animals exhibit a lack of smooth locomotion with frequent abrupt pauses before and after a touch on the head; as coiler if animals are observed in a coiled position (Supplementary Movie 1); as paralyzed if animals exhibit paralysis and fail to reverse upon light touch of the head with a platinum wire; and as 'wild-type locomotion' if animals exhibit normal sinusoidal locomotion and/or a normal backing response upon light touch on the head with a platinum wire. Coiled animals were not scored as paralyzed despite sometimes not responding to light touch. The *C. elegans* and *Pristionchus pacificus* dose-response assays presented in the structure activity analysis of Fig. 3c were conducted with animals on solid media with the indicated compound. After 1 h of being on solid media, animals were scored as 'Punc', 'coiled', 'paralyzed' or 'wild-type phenotype' and the presented data are the EC50 of animals demonstrating any motor phenotype.

### C. elegans 3-day development assays

*C. elegans* larval development assays were conducted in 96-well clear flatbottom plates. ~20 L1 larvae in 10 μL of M9 buffer were pipetted into each test wells containing 40 μL of Nematode Growth Media (NGM)[65] media supplemented with HB101 *E. coli* with the desired test compound (+0.6% dimethyl sulfoxide (DMSO) as the chemical solvent). Plates were wrapped in three layers of brown paper towels soaked with water. After either 3 days of incubation the number of *C. elegans* animals of different larval stages were recorded using a Leica MZ75 stereomicroscope.

### Radioligand competition assays for evaluation of phenylpiperazine VAChT affinity

Rat VAChT $K_i$ data are the $K_i$ of nemacol analog as determined from competitive radioligand [³H]Vesamicol binding assays using [³H]Vesamicol and PC12 cells expressing ratVAChT. [³H]Vesamicol was procured from PerkinElmer (PerkinElmer LAS GmbH, Rodgau, Germany; product ID: AH5183 (L-[Piperidinyl-3,4-3H]-(Vesamicol), $A_m$ = 1591 GBq/mmol)). Stably transfected PC12 cells expressing ratVAChT were shared by Dr. Ali Roghani (Texas Tech University, Lubbock, TX, USA) and radioligand competition experiments were conducted using standard protocols[66,67].

### *Dirofilaria immitis* culture and small-molecule assays

We conducted experiments with the Missouri isolate of *Dirofilaria immitis* in two labs. In the first set of experiments, shown as part of Fig. 3c, *D. immitis* microfilariae and larval stage 3 (L3) worms were assayed in the laboratories of Bayer Animal Health GmbH (Monheim, Germany). For microfilariae immobility assays, approximately 250 freshly purified microfilariae were cultured in single wells of a 96-well microtiter plate containing supplemented RPMI 1640 medium[68,69]. Compounds were added in the following concentrations: 50 μM, 10 μM, 2 μM, 0.4 μM, 0.08 μM, 0.016 μM and 0.0032 μM. Microfilariae exposed to medium substituted with 1% DMSO were used as negative controls. Motility of microfilariae was evaluated after 72 h of drug exposure using an image-based approach—DiroImager, developed by Bayer Technology Services. As described in detail in refs. [68,69], the DiroImager is an automated high-throughput platform that allows for high-resolution optical imaging of an entire 96-well microtiter plate. Data are reported as the EC50 (μM) calculated from the tested concentration series.

For the *Dirofilaria immitis* larval development assays, freshly isolated L3s were cultured in wells of a 96-well microtiter plate with 10 L3s per well. All wells contained supplemented RPMI 1640 medium[68,69] and a test compound at one of the following concentrations: 10 μM, 2 μM, 0.4 μM, 0.08 μM, 0.016 μM and 0.0032 μM. L3s exposed to DMSO only (1%) were used as negative controls. All drug concentrations were tested in duplicate and drug effects were evaluated after 72 h of incubation, after which DMSO-only controls are L4s and the shed L3 cuticles are evident in the wells. Compounds that have dramatic acute effects retard or arrest the growth of the nematodes and they remain L3s, while other compounds have less severe effects and the nematodes grow to the L4 stage. Motility was scored for each sample in the following standard manner: The 20 worms total from the two duplicate sample wells are considered. If only one of the 20 worms moves (as detected by pixel displacement with the DiroImager), the score for that sample is considered as 5% motility. Data on a given day were considered as valid only if the DMSO-only controls exhibited at least 90% motility. Data is then reported as the EC50 (μM) calculated from the tested concentration series.

We also conducted *Dirofilaria immitis* experiments in the lab of M. Zamanian (Univeristy of Wisconsin). For this experiment, shown in Supplementary Fig. 6, Microfilaremic blood was obtained from the NIH/NIAID Filariasis Research Reagent Resource Center (FR3)[70]. Blood was drawn and shipped approximately 24 h before use. Upon arrival, blood was warmed to 37 °C then combined with a 0.85% sodium chloride and 0.2% saponin solution in a 1:11 volumetric ratio and incubated in a 37 °C water bath for 15 min. The hemolyzed solution was passed through a 25 mm 5.0 μm pore size syringe filter. The used filter disks were transferred to petri dishes filled with RPMI 1640 culture media with 1% penicillin/streptomycin (0.1 mg/mL) and incubated at 37 °C for 2 h while microfilariae (mf) separated from the disks. Disks were discarded and mf titered to 5 mf/μL.

Aliquots of titered mf were incubated on a heating block at 60 °C for 1 h to produce heat killed positive controls. One μL of 100× drug was aliquoted to each well of a 96-well plate; 100 μL of live mf were added to treatment wells and 100 μL of heat killed mf were added to positive control wells (for a total of 500 mf/well). Plates were sealed with a breathable plate cover and maintained at 37 °C and 5% atmospheric $CO_2$. Videos of the plates were taken every 24 h for 72 h using an ImageXpress Nano (4×, 10 frames per well). A full protocol for the imaging process can be found here: https://doi.org/10.21203/rs.3.pex-1916/v2. After acquiring the 72-h timepoint video, viability staining was performed using the CellTox Green kit (Promega); a full protocol for this procedure can also be found at the previous link. Image analysis and the subsequent measurement of optical flow and fluorescence was performed using wrmXpress v1.3.02[71].

### Haemonchus contortus culture and small-molecule assays

The INVENesis Migration Trap Assay (MTA), previously described in ref. [72], measures the effect of compounds on the third larval (L3) stage of *Haemonchus contortus* nematodes (the susceptible isolate, 'Weybridge', UK). Approximately 300 L3s were exposed to the drug or control treatments in each well of a 96-well plate. In all experiments, non exsheated L3s were exposed for 24 h to the compound in a solution of 1.5% of DMSO and 0.00425% of Tween 20. Larvae were then transferred to a migration plate, which is a 96-well plate allowing *H. contortus* L3s to migrate from a deposit area to a trap area through a corridor. The mobility of the worms in the trap area is monitored over a 21 min time window by an automated data acquisition system equipped with a Basler acA2000-50gm camera with which pixel displacement is measured. The effect of compounds is expressed as a percent reduction of motility compared to negative controls. Nemacol-53 was tested in three replicates at the following concentrations: 120 μM, 60 μM, 30 μM, 15 μM, 7.5 μM, 3.75 μM, 1.875 μM, 0.9375 μM, and 0 μM. Ivermectin was tested in three replicates at the following concentrations: 20 μM, 5 μM, 1.25 μM, 0.3125 μM, 0.078125 μM, 0.0195 μM, 0.0049 μM, 0.0012 μM and 0 μM alone and along with the estimated dose corresponding to the EC30 of Nemacol-53.

### EMB-8 disruption

To disrupt the *C. elegans* cytochrome P450 oxidoreductase EMB-8, whose loss-of-function phenotype is embryonic lethality, we used the MJ69 strain harbouring the *emb-8(hc69)* temperature-sensitive allele in combination with RNAi targeting *emb-8* transcript degradation as done previously[73]. Embryonic lethality is avoided by growing MJ69 at 15 °C, obtaining synchronized L1s, placing the L1 animals on *emb-8(RNAi)*-inducing bacterial food (in the background of the HT115 E. coli strain), then shifting the temperature to 25 °C. Wild-type controls are treated the same way and grown on L4440 mock-RNAi-inducing bacterial food. The RNAi is used to increase the reduction of function of *emb-8*. The RNAi-by-feeding approach has been previously described[74]. Briefly, RNAi-inducing plates were prepared by growing liquid bacterial cultures overnight in the presence of 100 μg/mL ampicillin to

saturation. The following day, 6 cm MYOB agar plates with final concentrations of 100 μg/mL carbenicilin and 1 mM IPTG were seeded with 250 μL of the bacterial culture and dried at room temperature overnight. Worms were allowed to feed for 2 days before being transferred to experimental 'drug' plates. All 'drug' plates contained final concentrations of 100 μg/mL carbenicilin and 1 mM IPTG and were seeded with the *emb-8(RNAi)* and *L4440(RNAi)* control bacteria. Samples were then analyzed for phenotype as described above or for 'drug' metabolites, described in another section.

### High-performance liquid chromatography coupled to a diode array detector (HPLC-DAD) analyses of compound metabolism

Young adult worms were prepared on RNAi-inducing plates as described above. Adult worms were then incubated on drug plates for 6 h at 25 °C unless otherwise noted with 2000 worms per plate. After incubation, worms were washed off the agar plates and suspended in M9 buffer containing 0.5% gelatin[75]. After three washes in gelatin-M9, 500 μL of worm suspension was added to each well of Pall ACropPrep 96-well filter plates (0.45 μm GHP membrane, 1 mL well volume). The buffer was drained from the wells by vacuum and the worms were resuspended in 50 μL of M9 buffer and frozen at −80 °C. The samples were later lysed by adding 35 μl of a 2× lysis solution (100 mM KCl, 20 mM Tris, pH 8.3, 0.4% SDS, 120 μg ml$^{-1}$ proteinase K) to each tube at 56 °C for 1 h. Prior to HPLC coupled to a diode array detector (HPLC-DAD) analyses analysis, 70 μL of acetonitrile was added to the lysates. The samples were then mixed by vortexing for 10 s, and centrifuged at 17,949×g for 2 min. After centrifugation, 100 μL of the lysate was injected onto a 4.6 × 150 mm Zorbax SB-38 column (5-μm particle size) and eluted with solvent and flow rate gradients over 8.65 min. UV−Vis absorbance was measured every 2 nm between 190 and 602 nm. Using MATLAB (The MathWorks), absorbance intensity values were converted to three-dimensional heat-mapped chromatograms. A sample of 5 nmol pure Nemacol-1 was processed prior to worm samples to determine the compounds elution time and absorbance spectrum. All HPLC was performed a using an HP 1050 system equipped with an autosampler, vacuum degasser, and a variable wavelength diode-array detector. The solvent and flow rate gradients are indicated in the chart below. Data analysis was done using HP Chemstation software. The area under the curve (AUC) was automatically integrated by the software for quantification of Nemacol-1 absorbance peaks. For each sample, the ratio of parent and individual compound metabolite (M1, M2 and M3) AUC to the total AUC for all compound related peaks (parent, M1, M2 and M3) were calculated. As an example, for the calculation of the parent compound ratio, the AUC for the parent compound was divided by the total area under the curve for the parent compound and all metabolite related peaks.

### HPLC solvent and flow rate gradients

| Time (min) | Flow rate (ml/min) | Solvent A (%) | Solvent B (%) | High Pressure limit |
|---|---|---|---|---|
| 0.00 | 1.3 | 85.0 | 15 | 400 |
| 4.49 | 1.3 | 43.5 | 56.5 | 400 |
| 4.50 | 1.0 | 43.5 | 56.5 | 400 |
| 5.54 | 1.0 | 42.0 | 58.0 | 400 |
| 6.54 | 2.0 | 30.0 | 70.0 | 400 |
| 8.04 | 3.0 | 0.0 | 100.0 | 400 |
| 8.34 | 3.0 | 0.0 | 100.0 | 400 |
| 8.35 | 2.0 | 85.0 | 15.0 | 400 |

### Statistical analyses

Unpaired one- or two-sided *t* tests, or Chi-square tests were conducted between control and treatment groups where appropriate and as

indicated in figure legends. In Fig. 4b, the *P* values were calculated using the extra sum-of-squares F-tests comparing EC50 curves generated for dose-response data in GraphPad Prism (version 9.3.1). In Fig. 2a, significance was calculated using Fisher's method combining Fisher Exact Test values comparing to the wild-type response within each biological replicate. In Fig. 3a, the *P* value (*P* < E-15) was calculated using Fisher's Method with extra sum-of-squares F test comparing the binding curve of (−)Vesamicol to the binding curve of nemacol-1 conducted in parallel (on the same day with the same cell preparation).

## Commercial sources of nemacol phenylpiperazines

Nemacol analogs were procured from several commercial sources as dry compound. Molecules were sourced from Chembridge Inc., Enamine and OTAVA chemicals, Ltd. Vesamicol was purchased from Sigma-Aldrich as (±)-Vesamicol hydrochloride (product ID V100).

## Chemistry-general considerations

Unless otherwise stated, all reactions were carried out under an atmosphere of dry argon, using glassware that was either oven (120 °C) or flame-dried. Work-up and isolation of compounds was performed using standard benchtop techniques. All commercial reagents were purchased from chemical suppliers (Sigma-Aldrich, Combi-Blocks, or Alfa Aesar) and used without further purification. Dry solvents were obtained using standard procedures (dichloromethane and acetonitrile were distilled over calcium hydride). Reactions were monitored using thin-layer chromatography (TLC) on EMD Silica Gel 60 F254 plates. Visualization was performed under UV light (254 nm) or using potassium permanganate ($KMnO_4$) stain. Flash column chromatography was performed on Siliaflash P60 40-63 μm silica gel purchased from Silicycle. NMR characterization data were obtained at 293 K on a Varian Mercury 300 MHz, Varian Mercury 400 MHz, Bruker Advance III 400 MHz, Agilent DD2 500 MHz equipped with a 5 mm Xses cold probe or Agilent DD2 600 MHz. [1]H spectra were referenced to the residual solvent signal ($CDCl_3$ = 7.26 ppm, DMSO-$d_6$ = 2.50 ppm). [13]C{[1]H} spectra were referenced to the residual solvent signal ($CDCl_3$ = 77.16 ppm, DMSO-$d_6$ = 39.52 ppm). Data for [1]H NMR are reported as follows: chemical shift (δ ppm), multiplicity (s = singlet, d = doublet, t = triplet, q = quartet, m = multiplet), coupling constant (Hz), integration. NMR spectra were recorded at the University of Toronto Department of Chemistry NMR facility (http://www.chem.utoronto.ca/facilities/nmr/nmr.html). Infrared spectra were recorded on a PerkinElmer Spectrum 100 instrument equipped with a single-bounce diamond/ZnSe ATR accessory in the solid state and are reported in wavenumber (cm$^{-1}$) units. High-resolution mass spectra (HRMS) were recorded at the Advanced Instrumentation for Molecular Structure (AIMS) in the Department of Chemistry at the University of Toronto (https://www.chem.utoronto.ca/chemistry/AIMS.php).

## Chemistry-general procedure A

(4.0 mmol)   (4.0 equiv)   1a–f

The procedure was modified from literature[76]. To a round-bottom flask at room temperature were added piperazine (1.4 g, 16 mmol, 4.0 equiv), potassium carbonate (1.1 g, 8.0 mmol, 2.0 equiv), dimethyl sulfoxide (4.7 mL) then the corresponding fluorobenzene derivative (4.0 mmol, 1.0 equiv) and the mixture was stirred at 100 °C. Once the reaction was complete as indicated by TLC (approximately 22 h), the reaction mixture was diluted with ethyl acetate, transferred to a separating funnel and washed with water (three times) then saturated aqueous sodium chloride. The organic phase was dried with $Na_2SO_4$,

filtered, then concentrated on a rotary evaporator and the substituted 1-phenylpiperazine (**1a–f**) was obtained. The crude solid was used in the next step without further purification.

## Chemistry-general procedure B

To a round-bottom flask were added **1** (0.50 mmol, 1.0 equiv), dry acetonitrile (1.0 mL), triethylamine (0.14 mL, 1.0 mmol, 2.0 equiv) then the alkyl halide or toluenesulfonate (0.50 mmol, 1.0 equiv) and the mixture was stirred at reflux until completion as indicated by TLC (~24 h). The mixture was diluted with ethyl acetate, transferred to a separating funnel and washed with water (three times) then saturated aqueous sodium chloride. The organic phases were combined and dried with $Na_2SO_4$, filtered, and concentrated on a rotary evaporator. The residue was purified by column chromatography with the indicated eluent and the alkylated arylpiperazine (**2**) was obtained.

## Chemistry-general procedure C

(0.80 mmol)   3a

The procedure was modified from literature[77]. To a round-bottom flask were added cyclopentylmethanol (0.087 mL, 0.80 mmol, 1.0 equiv) and dry dichloromethane (1.6 mL) and the flask was submerged in an ice-water bath. Triethylamine (0.17 mL, 1.2 mmol, 1.5 equiv) was added to the flask followed by 4-(dimethylamino)pyridine (48.9 mg, 0.40 mmol, 0.50 equiv) then *p*-toluenesulfonyl chloride (168 mg, 0.88 mmol, 1.1 equiv) and the mixture was warmed to room temperature and stirred until completion as indicated by TLC (~3 h). Dichloromethane and water were added, then the mixture was transferred to a separating funnel and the layers were separated. The organic phase was washed with water (two times) followed by saturated aqueous sodium chloride, then dried over $Na_2SO_4$, filtered, and concentrated on a rotary evaporator. The residue was purified by column chromatography (19/1 v/v pentanes/ethyl acetate) and cyclopentylmethyl 4-methylbenzenesulfonate (**3a**, 59% yield) was obtained as a colourless oil. The spectral data were in accordance with literature.

## Characterization data for products
**4-(4-(cyclohexylmethyl)piperazin-1-yl)benzonitrile (2a, Nemacol-24).**

Synthesized according to general procedures A and B from 4-fluorobenzonitrile and (bromomethyl)cyclohexane. Compound **2a** was isolated as a bright yellow solid (MP = 61–63 °C). **[1]H NMR (CDCl₃, 500 MHz):** δ 7.46 (d, *J* = 9.0 Hz, 2H), 6.83 (d, *J* = 9.0 Hz, 2H), 3.32–3.28

(m, 4H), 2.52–2.48 (m, 4H), 2.15 (d, $J$ = 7.2 Hz, 2H), 1.78 (dddd, $J$ = 14.1, 4.9, 3.5, 1.5 Hz, 2H), 1.74–1.63 (m, 3H), 1.50 (ttt, $J$ = 10.8, 7.2, 3.5 Hz, 1H), 1.28–1.11 (m, 3H), 0.93–0.83 (m, 2H). **$^{13}$C NMR (CDCl$_3$, 125 MHz):** δ 153.6, 133.5, 120.2, 114.1, 99.6, 65.6, 53.3, 47.2, 35.1, 31.9, 26.9, 26.2. **IR (neat):** 2922, 2849, 2780, 2216, 1599, 1515, 1447, 1247, 1177, 818. **HRMS (DART):** calc. for $C_{18}H_{26}N_3$ 284.2127 [M + H]$^+$, found 284.2132.

**1-(cyclohexylmethyl)−4-(2-methyl-4-nitrophenyl)piperazine (2b, Nemacol-23).**

Synthesized according to general procedures A and B from 1-fluoro-2-methyl-4-nitrobenzene and (bromomethyl)cyclohexane. Compound **2b** was isolated as a bright yellow solid (MP = 52–54 °C). **$^1$H NMR (CDCl$_3$, 400 MHz):** δ 8.05–8.00 (m, 2H), 6.98 (dt, $J$ = 8.5, 1.1 Hz, 1H), 3.04 (t, $J$ = 4.8 Hz, 4H), 2.61–2.52 (m, 4H), 2.35 (s, 3H), 2.20 (d, $J$ = 7.1 Hz, 2H), 1.85–1.64 (m, 5H), 1.59–1.47 (m, 1H), 1.32–1.14 (m, 3H), 0.90 (q, $J$ = 12.5 Hz, 2H). **$^{13}$C NMR (CDCl$_3$, 125 MHz):** δ 157.7, 142.3, 132.2, 126.8, 122.8, 118.2, 65.7, 53.8, 51.2, 35.1, 32.0, 26.9, 26.3, 19.0. **IR (neat):** 2918, 2842, 2810, 1583, 1504, 1447, 1328, 1228, 1011, 893. **HRMS (DART):** calc. for $C_{18}H_{28}N_3O_2$ 318.2182 [M + H]$^+$, found 318.2178.

**1-(cyclohexylmethyl)−4-(3-methyl-4-nitrophenyl)piperazine (2c, Nemacol-25).**

Synthesized according to general procedures A and B from 4-fluoro-2-methyl-1-nitrobenzene and (bromomethyl)cyclohexane. Compound **2c** was isolated as a bright yellow solid (MP = 75–77 °C). **$^1$H NMR (CDCl$_3$, 500 MHz):** δ 8.06 (d, $J$ = 9.3 Hz, 1H), 6.68 (dd, $J$ = 9.3, 2.9 Hz, 1H), 6.61 (d, $J$ = 2.9 Hz, 1H), 3.39–3.35 (m, 4H), 2.62 (s, 3H), 2.53–2.49 (m, 4H), 2.16 (d, $J$ = 7.2 Hz, 2H), 1.82–1.75 (m, 2H), 1.75–1.64 (m, 3H), 1.50 (ttt, $J$ = 10.8, 7.2, 3.5 Hz, 1H), 1.29–1.12 (m, 3H), 0.94–0.83 (m, 2H). **$^{13}$C NMR (CDCl$_3$, 125 MHz):** δ 154.0, 139.0, 137.3, 127.9, 116.2, 111.2, 65.6, 53.3, 47.1, 35.1, 32.0, 26.9, 26.2, 22.8. **IR (neat):** 2920, 2843, 1638, 1600, 1533, 1249, 1226, 1001, 925, 822. **HRMS (DART):** calc. for $C_{18}H_{28}N_3O_2$ 318.2182 [M + H]$^+$, found 318.2189.

**1-(cyclohexylmethyl)−4-(pyridin-4-yl)piperazine (2d, Nemacol-22).**

Synthesized according to general procedures A and B from 4-chloropyridine hydrochloride and (bromomethyl)cyclohexane. Compound **2d** was isolated as a white solid (MP = 66–70 °C). **$^1$H NMR (DMSO-$d_6$, 500 MHz):** δ 8.17 (d, $J$ = 6.1 Hz, 2H), 6.92 (d, $J$ = 6.1 Hz, 2H), 3.40 (t, $J$ = 5.1 Hz, 4H), 2.43 (t, $J$ = 5.1 Hz, 4H), 2.12 (d, $J$ = 7.2 Hz, 2H), 1.76–1.70 (m, 2H), 1.69–1.58 (m, 3H), 1.50 (ttt, $J$ = 10.9, 7.3, 3.4 Hz, 1H), 1.26–1.07 (m, 3H), 0.89–0.77 (m, 2H). **$^{13}$C NMR (DMSO-$d_6$, 125 MHz):** δ

155.2, 146.3, 108.0, 64.6, 52.6, 45.5, 34.2, 31.2, 26.4, 25.5. **IR (neat):** 2919, 2849, 1638, 1600, 1533, 1249, 1132, 1001, 989, 805. **HRMS (DART):** calc. for $C_{16}H_{26}N_3$ 260.21267 [M + H]$^+$, found 260.2120.

**4-(4-(cyclopentylmethyl)piperazin-1-yl)benzonitrile (2e, Nemacol-27).**

Synthesized according to general procedures A, B and C from 4-fluorobenzonitrile and **3a**. Compound **2e** was isolated as a light orange solid (MP = 49–51 °C). **$^1$H NMR (CDCl$_3$, 500 MHz):** δ 7.47 (d, $J$ = 9.0 Hz, 2H), 6.84 (d, $J$ = 9.0 Hz, 2H), 3.34–3.29 (m, 4H), 2.59–2.52 (m, 4H), 2.30 (d, $J$ = 7.3 Hz, 2H), 2.08 (hept, $J$ = 7.3 Hz, 1H), 1.80–1.71 (m, 2H), 1.65–1.48 (m, 4H), 1.26–1.16 (m, 2H). **$^{13}$C NMR (CDCl$_3$, 125 MHz):** δ 153.6, 133.6, 120.3, 114.1, 100.0, 64.5, 53.1, 47.2, 37.2, 31.5, 25.3. **IR (neat):** 2947, 2936, 2864, 2816, 2211, 1601, 1513, 1448, 1166, 921. **HRMS (DART):** calc. for $C_{17}H_{24}N_3$ 270.19702 [M + H]$^+$, found 270.1968.

**1-(cyclopropylmethyl)−4-(4-nitrophenyl)piperazine (2 f, Nemacol-28).**

Synthesized according to general procedures A and B from 1-fluoro-4-nitrobenzene and (bromomethyl)cyclopropane. Compound **2f** was isolated as a dark orange solid. The characterization data are in accordance with literature[78]. **$^1$H NMR (CDCl$_3$, 500 MHz):** δ 8.12 (d, $J$ = 9.4 Hz, 2H), 6.82 (d, $J$ = 9.4 Hz, 2H), 3.46 (t, $J$ = 5.2 Hz, 4H), 2.68 (t, $J$ = 5.2 Hz, 4H), 2.32 (d, $J$ = 6.6 Hz, 2H), 0.90 (ttt, $J$ = 8.0, 6.6, 4.9 Hz, 1H), 0.58–0.54 (m, 2H), 0.16–0.12 (m, 2H). **$^{13}$C NMR (CDCl$_3$, 125 MHz):** δ 155.0, 138.5, 126.1, 112.7, 63.7, 52.8, 47.1, 8.4, 4.1.

### Reporting summary

Further information on research design is available in the Nature Portfolio Reporting Summary linked to this article.

## Data availability

Original data for all analyses presented are included in the Source data file. Sequences used herein include the following (NCBI reference sequence identifiers or otherwise stated): human NP_003046.2; swine (*Sus scrofa*): XP_013838900.2; cattle (*Bos taurus*): XP_002699016.1; sheep (*Ovis aries*): XP_027818269; mouse (*Mus Musculus*): NP_068358.2; rat (*Rattus norvegicus*): NP_113851.1; zebrafish (*Danio rerio*): NP_001071018.1; *Trichuris trichiura*: CDW52212.1; *Ancylostoma duodenale*: KIH66835.1; *Necator americanus*: XP_013297134.1; *Onchocerca volvulus*: A0A2K6VZC1 (UniProt ID) *Ascaris suum*: AgB02_g088_t01 (WormBase ParaSite transcript ID); *Dirofilaria immitis*: nDi.2.2.2.t09212 (WormBase ParaSite transcript ID); *Haemonchus contortus*: A0A7I4YIM0 (UniProt ID); *C. elegans*: NP_001379838.1. Source data are provided with this paper.

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

## Acknowledgements

We are grateful for the strains given to us by the *Caenorhabditis* Genetics Center (University of Minnesota) and James Rand (Oklahoma Center for Neuroscience). Christopher Evans and Andrew Moorhead of the NIH/NIAID Filariasis Research Reagent Resource Center (FR3) provided valuable advice on *Dirofilaria* purification and provided some parasite material. Some data analysis was performed using the compute resources and assistance of the UW-Madison Center For High Throughput Computing (CHTC) (Dept of Computer Sciences). The following grants supported this work: CIHR grant 313296 (P.J.R.), CIHR grant 173448 (P.J.R.), Canada Research Chair program (P.J.R.), NSERC grant RGPIN-2020-04168 (M.L.) and an NIH/NIAID grant R01 AI151171 (M.Z.).

## Author contributions

The work was conceptualized by P.J.R., S.H., and J.P. The methodology was developed by S.H., J.P., T.S., R.J.B., A.R.B., M.L., D.K., W.D.-C., L.R., A.V., K.T.R., M.Z. and P.J.R. The wetlab investigation was performed by S.H., J.P., T.S., R.J.B., A.R.B., J.C., D.K., W.D.-C., L.R., A.V., and K.T.R. The figures were composed by P.J.R., S.H., and J.P. Funding was acquired by P.J.R. and M.Z. The project was led and administered by P.J.R. Traineed supervision was provided by P.J.R., M.L., D.K., W.D.-C., P.B., and M.Z. The original draft was written by S.H., J.P., and P.J.R. and submitted and revised versions were written by P.J.R., S.H., J.P., R.J.B., M.L., and P.B.

## Competing interests

Mention of trade names or commercial products in this publication is solely for the purpose of providing specific information and does not imply recommendation or endorsement by any author or affiliate organization. S.H., J.P., A.R.B., R.J.B., D.K., M.L. and P.J.R. have an issued U.S. patent covering the Nemacol scaffold (U.S. patent # 11,364,234) and additional patents pending related to the Nemacol scaffold. All molecules described herein can be used without restriction for academic purposes. The remaining authors declare no competing interests.
