## [Peer Review File · Nature Communications]

Reviewers' Comments:

Reviewer #1:

Remarks to the Author:

This research is an important contribution in the identification of new anthelmintics acting through a novel mechanism of action, viz, the vesicular acetylcholine transporter of nematodes. The results are clearly presented and fully support the conclusions. No additional evidence is required. The methodology is sound and meets the standards expected in the field. However, where data are significantly different from controls or between experimental results, this should be clearly stated, both in the text and figure legends. I have the following comments/queries:

line 78: spiroindoline

line 97 and elsewhere: Fig 1A-D as capitals as used in figures

lines 101/2: indicate in text this is significant

lines 126 & 131: indicate level of significance

line 193: perdue? - perhaps rephrase

209: not in references

215: need to explain SwissADME and 218: synthetic accessibility score

line 247: how quickly do nematodes cross *C. elegans* cuticle? Any idea of concentration in worm?

251: was this the lowest concentration of DMSO that could be used in these experiments? This concentration can have direct effect on *C. elegans* (Toxicol Reports 8:1240-7 (2021))

264 & 447: are these L4+1?

278: well and 295: were

433: explain -reduced the drug activity by 5% -

490: References - need checking; some have all the authors, others just the first author; some are duplicated, 27/31; 32/36

Figure 1B & D: was there full recovery after 24hrs?, comment in legend

Difficult to see colour difference between $p < 0.05$ and 0.001 , suggest change colour coding or use *, **, ***

The horizontal bar under B and C looks to be incorrect. I'd assume white was 0%

Comment on why use L4 in figure 1 and adult in figure 2, were these L4+1?

729: (C) should be (B) and add (C) after trichlorfon

Mention use of UNC-17 in figure 2 legend title

Figure 3C needs to be enlarged

747/8: how long were animals exposed to drugs, add to text?

750: in figure 3C, these values referred to as LC50 but in legend as EC50, both correct but should be consistent

In figure 3C, comment why some compounds were not tested

Figure 4: in (B), indicate if the EC50s are significantly different from control

Figure S1: add EMB-8 to title

Figure S2: under A, I'd assume the white was 0% paralysed and in (B), indicate which points are significantly different from control

Reviewer #2:

Remarks to the Author:

The primitive formatting available with the fill-in boxes on the submission website makes my review harder to read. Accordingly, I am submitting my review as an attached pdf.

Reviewer #3:

Remarks to the Author:

In the present study, the authors investigated the anthelmintic activities of Nemaicol, a nitrophenyl-piperazine scaffold. Starting from 4 compounds selected from a screen performed in another study, they use *C. elegans* as a model to decipher the mode of action of these drugs, highlighting the vesicular acetylcholine transporter UNC-17 as the main target. Nemaicol-1 was investigated in more details, including the role of drug metabolizing CytP450 in its transient activity on the worms. Subsequently the authors investigated Nemaicol analogs for acute motor tests in *C. elegans*, *P. pacificus* and the Dog heartworm *D. immitis*. Finally, the authors tested combinations of Nemaicol with the widely used anthelmintic Ivermectin and reported a promising synergic effect.

This paper is of great interest as it reveals a novel class of anthelmintic for which there is an urgent need. Even though the present content lays a strong basis for an attractive and original article, I would recommend important modifications to the manuscript.

First: As mentioned by the authors (line 94) the four molecules which paved the way for this study have been identified from a previous screen (ref 10). In the mentioned paper, the authors took care of investigating the activity of the compounds on a range of parasitic nematode species as well as off-target species, thus providing critical information on both the spectrum of activity and the potential toxicity for vertebrates. In the present work, the authors did not refer to the molecule's nomenclature used in this "reference" paper. Therefore, in order to support their claim concerning the potential use of Nemaicol as an anthelmintic treatment, the authors should provide for each compound the information concerning both the activity observed on the other parasitic nematode species as well as the data concerning the off-target species. Note that the authors mentioned in the discussion part (line 191) that "assays performed against multiple parasitic nematode species over an equivalent timespan indicate that Nemaicol's effects can perdure in parasites", however they did not mention a corresponding reference and in the present study there is no data supporting this claim.

Second: The work on *Dirofilaria immitis* is incomplete

Based on the main objective of the paper, there are too few attempts to transpose the results from *C. elegans* to the parasite species.

For *C. elegans*, the authors described in details the different phenotypes (over time) associated with the drug exposure. Here the same information should be provided for *D. immitis*.

The description of the screening approach on *D. immitis* is minimalistic, there is no information about the precise parameters that have been monitored using the imaging approach.

Apart from the EC50 values (without SE information...) there is no data nor statistical analysis provided. In contrast with the *C. elegans* experiments, the authors did not clearly indicate the number of independent trials (independent batches of worms) used for *D. immitis* experiments.

Concerning the L3 (for which surprisingly the results have been reported as L4 in fig.4C), the authors mentioned a developmental assay whereas they performed a motility assay. The sentence (lines 432-433) should be reformulated as we could only hardly guess what the authors meant while describing the test.

On *D. immitis*, the scoring was performed after 72h exposure whereas for *C. elegans*, three among the four tested drugs (in the first part of the manuscript), either lost or had a reduced activity after 4h. This should be explained.

Finally, whereas the search for synergy identified a positive interaction between Nemaicol-1 and IVM in *C. elegans*, could the authors justify why there was no attempt to transpose this important result to *D. immitis*, knowing that Macrocyclic lactones are used for the treatment of this parasite. If such "synergy" could be confirmed in a parasitic nematode species, this would represent an important proof of concept thus increasing the impact of the paper.

Third: search for synergy

Whereas the discovery of drug combinations leading to a lethal phenotype is of major interest, this part needs to be improved to further support the authors claim.

The authors should justify why they use a developmental assay (72h) instead of the motility assay. Indeed, this could be misleading as the authors decided to use Nemaicol-1 for which they demonstrated a transient activity (24h) and then provide an evidence that the same drug is able to kill some worms at 60µM after 72h...this observation should be mentioned in the first part of the manuscript.

Line 170-172 the authors state that "combination of IVM and Nematicol could yield effective killing of *C. elegans* at concentrations that had little effect on their own".

A "little effect" could be interpreted in many different ways. Unfortunately, Fig4A is very difficult to read and it is impossible to accurately attribute a value to the different purple nuances and distinguish potential additive effects, synergies or even absence of effect. The authors should provide (at least for a set of relevant combinations) a detailed statistical analysis to support their claim. Note that Fig4C without highlighting the zone of interest doesn't bring a lot of help for the reader.

The Fig4 B is much more convincing (however note that based on the dose-response curve, the EC50 values for IVM should be in nM and not in μ M) and lays a strong basis for the demonstration. The authors should provide the dose response curve for Nematicol-1 with fixed concentrations of IVM and it would be also of major interest to investigate the effect of Nematicol-1 on an IVM resistant *C. elegans* strain such as DA1316 mutant in order to mirror the approach presented in Fig4B

General comments: the authors should justify their choice concerning the focus they made on Nematicol-1, which harbours a transient activity, instead of Nematicol-2 for which there was no apparent drastic reduction of activity after 4h while sharing similar EC50 values with Nematicol-1 on *C. elegans*. Note that a test at 1440 min (as for Nematicol-1) would be very informative for Nematicol-2. Similarly, the authors should justify the selection they made for the drugs used in the acute tests performed on *D. immitis*. What was the reason for not investigating Nematicol-2 on the parasite whereas it appeared to more relevant than Nematicol-1 for the control of a parasite species that cannot be expelled from the host ?

For all the study, EC50 should be provided with their corresponding SE, indeed the variability of the results obtained with some compounds appears to be very important (Nematicol-3 and 4 for example) Dose response curves obtained during the acute tests should be provided at least as supplemental data.

The discussion is currently mainly focused on the potential industrial applications (including comments on results which are not presented in the present work...L211-223). Even though this represent an important aspect, the discussion part would largely benefit of comments and prospects about other important points raised by the present work (Nematicol detoxification, synergies, drug resistance management...)

Minor comments:

In Fig 2 and S2 the colour code has been inversed (i.e. with dark red indicating 0% paralysis and white indicating 100% paralysis)

Review of Manuscript NCOMMS-22-23586, by Harrington *et al.*

The vesicular acetylcholine transporter (VACHT) is an integral membrane protein required to transport acetylcholine into synaptic vesicles. It is the only known marker specific for cholinergic synapses, and loss of VACHT activity is lethal to animals that use acetylcholine as a neurotransmitter. These studies describe a new type of VACHT-binding compound, an inhibitor that could be the entry point for the development of useful tools to study VACHT pharmacology. Although the manuscript is focused on the development of anti-VACHT compounds for the treatment of human and animal nematode parasitic infections, there are other VACHT-related avenues of potential interest. The only previously known VACHT-binding compounds (vesamicol, spiroindolines) can not be used as PET tracers, because they also bind to sigma receptors. If a VACHT-specific Nematicol derivative could be developed, it could become a useful medical tool. In addition, human VACHT mutations are associated with a type of congenital myasthenic syndrome, a rare but debilitating disorder.

I believe that these studies are definitely worthy of publication. The manuscript is clearly written (except for the details of chemical syntheses), and the drug binding and drug analog studies are well done. However, there is one area which can be improved, namely the behavioral phenotyping.

The 3 phenotypic categories:

Jerky-Unc: This is a subjective descriptor, which can apply to many types of abnormal locomotion, from mild to severe. The stated criteria for “jerky-unc” are that “animals exhibit a lack of smooth locomotion with frequent abrupt stops and start before and after a touch on the head.” This confuses me – if the stops and starts occur both before and after a touch on the head, and the worm fails to reverse direction, this implies that the animals are touch-insensitive.

In Figure 3C, the terminology “*C. elegans* motor disruption” is used. In some parts of the Supplemental Data spreadsheet, the term “atypical locomotion” is used. I think that either one of these descriptors would be preferable to “jerky-unc.”

Paralyzed: One might predict that decreasing the release of an excitatory neuromuscular transmitter would lead to a flaccid paralysis, but this is not the case. Severe (but non-null) *unc-17* and *cha-1* mutants are not flaccid at all, but rather are quite uncoordinated, often tightly coiled (such as the upper worm in the video), and jerky (almost ratchet-like) when backing. Unfortunately, in the manuscript the terms paralysis and flaccid paralysis seem to be used interchangeably. (For a good example of a truly flaccid phenotype, look at *unc-52* homozygotes.) If in fact, Nematicol exposure produced a really flaccid phenotype, it would be unexpected and of great interest.

Coiled: This is an accepted descriptor, and is straightforward to score. However, the stated criteria for “paralysis” (“animals...fail to reverse upon light touch of the head with a platinum wire”) would also apply to tightly coiled animals.

It looks like every worm was assigned to one of these 3 phenotypes or wild-type. What about animals that didn't fit into any of the 4 phenotypes (lethargic animals, “loopy” Uncs, other types of non-jerky Uncs)? Also, do the 3 non-wild-type phenotypes represent a gradient of severity?

Phenotypic measurements, statistics, and plotting:

The color density plots in Figures 2B, 2C, 4A, and S2A are confusing. The authors perform statistical analysis on a large set of what are admittedly subjective descriptions, and plot the results in a confusing (to me, at least) way. This isn't necessary, because there's no need to provide detailed quantitative measurements of the effects of these drugs on behavior. For example, the first set of results demonstrates the *unc-17* mutants are hypersensitive to Nematicol; the second set of results demonstrates that *md414* mutants are partially resistant to Nematicol; and the third set of results demonstrates that Nematicol-1 can suppress the paralysis induced by AChE inhibitors. In each case,

quantitative details (e.g., precise EC50 values) are less important than the (qualitative) conclusion. For qualitative conclusions, comparative images of worms are better than graphs – readers can grasp the conclusions more easily and in less time. See, for example, Figures 4A, 5, and 6 from the Sluder paper.

=====

Lines 63-65: “The neurotransmitter acetylcholine (ACh) is the primary signalling molecule in **most** animals that triggers muscle contraction.”

Not in all animals – for example, insects use glutamate as the excitatory neuromuscular transmitter.

=====

Line 75: “...is inhibited by **vesamicol and by** the relatively novel spiroindoline scaffold”

=====

Lines 226-229: “VACHT mutant residues that confer resistance to vesamicol or the spiroindolines disrupt the ability of these molecules to interact with VACHT, but reduce acetylcholine interaction and lead to obvious locomotory defects in *C. elegans*.”

This statement is incorrect. According to the Sluder paper, the spiroindoline-resistance mutants they obtained “had no obvious phenotype as homozygotes.”

=====

Lines 229-231: “Missense mutations that alter the ability of VACHT inhibitors to interact with the transporter will likely confer a clear selective disadvantage to worms.”

This line of reasoning is flawed. Although animals carrying such VACHT/UNC-17 mutations may be at a selective disadvantage in the absence of the drug, in the presence of the drug they are quite likely to outcompete wild-type animals. This is why pesticide-resistance is such a problem in the field.

=====

Lines 284-403: This material is outside my field of expertise, and I am not able to comment on it.

=====

Minor Details:

Line 82: “...which is a phenotype shared by mutants of the CHA-1 **choline** acetyltransferase enzyme...”

Line 683: Change “Oklahoma (Center for Neuroscience)” to “(Oklahoma Center for Neuroscience)”

=====

References:

Reference #31 is a duplicate of reference #27. Reference #36 is a duplicate of #32. There may be other examples as well.

Reference #60 provides 4 authors and a publication year, but no title and no citation.

=====

Figures:

Figure 2A: The data cluster at 0 μ M Nemaicol is missing the data for *e327*, *e795*, and *md414*, all of which are significantly uncoordinated.

Figures 2B, 2C, S2A: the horizontal color-density scale is backwards.

Figures 2B, 2C, 4A, S2A: See the paragraph above on plotting, and the 2nd recommendation below.

=====

Video:

The tightly coiled animal appears quite similar to severe loss-of-function *unc-17* or *cha-1* mutants. The other two worms do not display normal locomotion, but I'm not sure I would describe their movement as jerky. In any event, they do not appear to be seriously incapacitated.

=====

Recommendations:

- Lines 225-231 (the 2nd, 3rd, and 4th sentences of the final Discussion paragraph) are factually incorrect, and should be eliminated.
- Figures 2B, 2C, 4A, and S2A should be eliminated, and replaced by images of worms at specific data points. See, for example, Figures 4A, 5, and 6 from the Sluder paper. **HOWEVER**, if this recommendation is unacceptable for some reason, my (reluctant) fallback suggestion for these figures is to make them at least less confusing by following the example used in Figure S3A, and entering a specific value in each cell.

Reviewed by James B. Rand, Ph.D.

Reviewer #1 (Remarks to the Author)

This research is an important contribution in the identification of new anthelmintics acting through a novel mechanism of action, viz, the vesicular acetylcholine transporter of nematodes. The results are clearly presented and fully support the conclusions. No additional evidence is required. The methodology is sound and meets the standards expected in the field.

Our Response: We thank the reviewer for their kind words and encouragement.

Reviewer #1- Comment 1: However, where data are significantly different from controls or between experimental results, this should be clearly stated, both in the text and figure legends.

Our Response: We have made the requested revisions. In addition to the original determinations of significance, we have added measures of significance to the following figure panels: Figures 1a-1e, 2c-2e, 3a, 3b, 4a, 4c, 4e, Supplemental Figures 1d, 2a, 2b, 7a, and 7b. We have made additional notes of important significant differences in the text where relevant.

Reviewer #1- Comment 2: line 78: spiroindoline

Our Response: We have corrected the spelling error- thank you for catching.

Reviewer #1- Comment 3: line 97 and elsewhere: Fig 1A-D as capitals as used in figures

Our Response: We have corrected notation to lower case throughout according to Nature Communications formatting.

Reviewer #1- Comment 4: lines 101/2: indicate in text this is significant

Our Response: We have made the requested changes and have altered the sentence to read as, 'Blocking drug-metabolizing cytochrome P450s by knocking down *C. elegans* cytochrome P450 reductase (EMB-8) antagonizes both the metabolism of Nematicol-1 ($p < 0.05$) the dissipation of these phenotypes ($p < 0.001$) (Fig. S1).'

Reviewer #1- Comment 5: lines 126 & 131: indicate level of significance

Our Response: We have made the requested changes as follows (changes underlined):

We find that 30 μ M Nematicol-1 yields nearly an identical 2.2-fold shift in the EC50 of trichlorfon paralysis compared to 250 μ M vesamicol representing an 8.3-fold shift in potency ($p < 1E-15$) (Fig. 2e).

We found that Nematicol-1 binds mammalian VACHT, but with 37-fold less affinity than vesamicol ($p < 1E-15$) (Fig. 3a).

Reviewer #1- Comment 6: line 193: perdue? - perhaps rephrase

Our Response: We have changed 'perdue' to 'persist'.

Reviewer #1- Comment 7: 209: not in references

Our Response: The reference in the original 209 line (Sluder et al, 2012) is in the references of the revised manuscript.

Reviewer #1- Comment 8: 215: need to explain SwissADME

Our Response: In the original and revised manuscript, we provide a reference that describes the SwissADME chemoinformatics tool, which is widely used (>1400 citations). To address the Reviewer's concern, we have added the following sentence 'SwissADME is a highly cited chemoinformatics tool that is widely used in the field⁵⁷.'

Reviewer #1- Comment 9: 218: synthetic accessibility score

Our Response: We have modified the phrase as follows (new word underlined): 'MW 453.53 with a SwissADME synthetic accessibility score of 4.23'. In addition, we write, 'The higher the synthetic accessibility score, the more difficult the synthesis^{57, 58} to explain what a synthetic accessibility score is. Furthermore, we added a few new columns to Figure 3c, including the synthetic accessibility score for each of the molecules listed.'

Reviewer #1- Comment 10: line 247: how quickly do nemacols cross *C. elegans* cuticle? Any idea of concentration in worm?

Our Response: Figure 1 reports the timing of phenotypic onset for several analogs. The earliest time point that we collected data for is 30 minutes, when at least for analog 1, we see robust coiling and atypical locomotion. These observations suggest that nemacol accumulates quickly within the tissues in the animal, and has therefore at least partially circumvented the xenobiotic barriers within that time frame. We have not measured concentration of nemacol in the worm. To acknowledge these points within the context of the results, we have added the following sentence at the end of the first paragraph of the results section: 'Effectiveness of the different analogs likely reflects their different rates of accumulation, metabolism, detoxification and target engagement.'

Reviewer #1- Comment 11: 251: was this the lowest concentration of DMSO that could be used in these experiments? This concentration can have direct effect on *C. elegans* (Toxicol Reports 8:1240-7 (2021)).

Our Response: Yes, 1% was the lowest concentration of DMSO that we used. It is the standard drug concentration that our lab (and others) uses in our chemical-genetic experiments and screens. For examples, see:

Kwok *et al.* 2006. A Small Molecule Screen in *C. elegans* Yields a New Calcium Channel Antagonist. *Nature*. 441, 91-95. PMID: 16672971

Burns *et al.* 2010. A Predictive Model for Drug Bioaccumulation and Bioactivity in *Caenorhabditis elegans*. *Nature Chemical Biology*, 6, p549-557 (doi: 10.1038/nchembio.380). PMID: 20512140

Burns *et al.* 2015. *Caenorhabditis elegans* is a useful model for Anthelmintic Discovery. *Nature Communications*. 6:7485 | DOI: 10.1038/ncomms8485. PMID: 26108372

Zwirchmayr *et al.* 2020. A robust and miniaturized screening platform to study natural products affecting metabolism and survival in *Caenorhabditis elegans*. *Sci Rep* 10, 12323 (2020). <https://doi.org/10.1038/s41598-020-69186-6>

We have previously performed (unpublished) dose response analyses with DMSO and see obvious effects of DMSO on the growth of worms when used above a concentration of 1.5%, but no obvious effects at 1%. Furthermore, all of our experiments are controlled relative to DMSO-only controls, and dose-response trends are observed, indicating that the effects we report are due to the small molecule and not the solvent.

Reviewer #1- Comment 12 264 & 447: are these L4+1?

Our Response: We have added clarity to these points by writing in the Worm Culture and Strains methods section that: 'Synchronized young adult animals are acquired by incubating synchronized L1s (first larval stage animals hatched overnight in M9 buffer (Lewis and Fleming JT, 1995) on plates with *E. coli* (strain OP50) food for 64-72 hours at 20°C.' Subsequent methods sections simply refer to 'synchronized young adults'.

Reviewer #1- Comment 13: 278: well and 295: were

Our Response: The errors have been corrected.

Reviewer #1- Comment 14: 433: explain -reduced the drug activity by 5% -

Our Response: We apologize for the error. The relevant sentence has been corrected to, 'Motility was scored for each sample in the following standard manner: The 20 worms total from the two duplicate sample wells are considered. If only one of the 20 worms moves (as detected by pixel displacement with the Dirolmager), the score for that sample is considered as 5% motility.'

Reviewer #1- Comment 15: 490: References - need checking; some have all the authors, others just the first author; some are duplicated, 27/31; 32/36

Our Response: We apologize for the error and have corrected the oversight in the revised manuscript.

Reviewer #1- Comment 16: Figure 1B & D: was there full recovery after 24hrs? Comment in legend.

Our Response: In the original submission, we presented data in Figure 1 for the 24 hour timepoint for only Nemaicol-1. In the revised manuscript, we present new 24 hour data for all of the analogs shown in Figure 1 (along with parallel new negative (DMSO-only) controls). At the 24 hour timepoint, *C. elegans* worms recover from all analogs except Nemaicol-2, where uncoordinated locomotion persists, suggesting that *C. elegans* metabolism of Nemaicol-2 is a little slower relative to the other three analogs.

Reviewer #1- Comment 17: Difficult to see colour difference between $p < 0.05$ and 0.001 , suggest change colour coding or use *, **, ***

Our Response: We have changed the colour scheme to more distinct colours.

Reviewer #1- Comment 18: The horizontal bar under B and C looks to be incorrect. I'd assume white was 0%.

Our Response: We apologize for the error. We have corrected the oversight in the revised manuscript (note that original Figures 2B and 2C are now revised Figures 2C and 2D).

Reviewer #1- Comment 19: Comment on why use L4 in figure 1 and adult in figure 2, were these L4+1?

Our Response: We apologize for the oversight. After consulting our notes (from Jacob Pyche's thesis in which the data was reported), the animals in Figure 1 were indeed young adults (as were the animals used in the new 24 hour timepoint data). The figure legend has been updated accordingly.

Reviewer #1- Comment 20: 729: (C) should be (B) and add (C) after trichlorfon

Our Response: We thank the reviewer for catching this error and have made the correction.

Reviewer #1- Comment 21: Mention use of UNC-17 in figure 2 legend title

Our Response: We have changed Figure 2's title to (changes underlined): Figure 2. Nemaicol induces phenotypes consistent with the inhibition of UNC-17 (VACHT).

Reviewer #1- Comment 22: Figure 3C needs to be enlarged

Our Response: We have nearly doubled the size of Figure 3C.

Reviewer #1- Comment 23: 747/8: how long were animals exposed to drugs, add to text?

Our Response: Animals were treated for one hour, which we have indicated in the revised legend.

Reviewer #1- Comment 24: 750: in figure 3C, these values referred to as LC50 but in legend as EC50, both correct but should be consistent.

Our Response: The assay reports the reduction of locomotion, so EC50 is the more appropriate term (despite the inference being that immobile worms are likely dead). We have changed the chart to EC50.

Reviewer #1- Comment 25: In figure 3C, comments why some compounds were not tested.

Our Response: With respect to the absent *P. pacificus* data, we simply ran out of compound at the time the assays were performed.

With respect to the absent *Dirofiliaria* data, Nemaicol-2 showed higher activity in our preliminary HEK293 cell-based assays (see our Burns et al., 2015 paper), so we elected not to pursue it in Kulke's assay.

With respect to the rat VAcHT affinity tests, we selected only a subset of molecules (based on EC50 information in other assays that we had data for at the time) because of the expensive nature of the assay. The VAcHT affinity test is expensive because it relies on [³H] vesamicol. We therefore selected only a subset of analogs to test based on activity data in *C. elegans* that we had in hand at the time. We also selected a diversity of structure. We highlight this point in the original and revised results section as follows, 'Next, we chose 12 diversely structured analogs that had good activity against *C. elegans* and investigated whether any might have weakened affinity for the rat VAcHT relative to vesamicol and Nemaicol-1.'

We also make a new note of this in the figure legend by writing, 'The grey cells lack data because of either insufficient material at the time the assays were performed, lethality in non-targeted systems, or because of the expense associated with the rat VAcHT binding assays.'

Reviewer #1- Comment 26: Figure 4: in (B), indicate if the EC50s are significantly different from control

Our Response: The EC50s and the respective differences of significance relative to the 'no Nemaicol-1' treatment for Figure 4b (called Figure 4c in the revision) are now listed on the right-hand side of the revised chart in Figure 4a.

In addition, we have modified the following sentence in the relevant results paragraph (relevant additions underlined): 'Indeed, *unc-17(e245)* demonstrated a sensitivity to ivermectin ($p < 1E-12$ relative to wild type's sensitivity to ivermectin) that was comparable to 30 μ M Nemaicol treatment ($p < 1E-15$ relative to wild type's sensitivity to ivermectin) (Fig. 4c).'

Reviewer #1- Comment 27: Figure S1: add EMB-8 to title

Our Response: We have changed the Figure title to, 'Disruption of the EMB-8 cytochrome p450 reductase suppresses *C. elegans*' ability to metabolize Nemaicol and the worms' ability to recover from Nemaicol-1-induced motor defects.' We have also added additional relevant experimental details in the Figure S1 legend and a new 'EMB-8 Disruption' methods section in the revised manuscript.

Reviewer #1- Comment 28: Figure S2: under A, I'd assume the white was 0% paralysed

Our Response: We apologize for the error and have corrected the oversight in the revised manuscript.

Reviewer #1- Comment 29: Figure S2: in (B), indicate which points are significantly different from control.

Our Response: The EC50s of the two curves in Figure S2b are significantly different ($p < 8.5E-5$) and have added that statistic to the figure panel as well as a description of the test in the legend.

Reviewer #2 (James B. Rand, Ph.D.)

The vesicular acetylcholine transporter (VACHT) is an integral membrane protein required to transport acetylcholine into synaptic vesicles. It is the only known marker specific for cholinergic synapses, and loss of VACHT activity is lethal to animals that use acetylcholine as a neurotransmitter. These studies describe a new type of VACHT-binding compound, an inhibitor that could be the entry point for the development of useful tools to study VACHT pharmacology. Although the manuscript is focused on the development of anti-VACHT compounds for the treatment of human and animal nematode parasitic infections, there are other VACHT-related avenues of potential interest. The only previously known VACHT-binding compounds (vesamicol, spiroindolines) can not be used as PET tracers, because they also bind to sigma receptors. If a VACHT-specific Nemacol derivative could be developed, it could become a useful medical tool. In addition, human VACHT mutations are associated with a type of congenital myasthenic syndrome, a rare but debilitating disorder.

I believe that these studies are definitely worthy of publication. The manuscript is clearly written (except for the details of chemical syntheses), and the drug binding and drug analog studies are well done.

Our Response: We thank Professor Rand for the kind words and valuable perspective. It is an honour to have him review our work.

Reviewer #2- Comment 1: However, there is one area which can be improved, namely the behavioral phenotyping. The 3 phenotypic categories: **Jerky-Unc:** This is a subjective descriptor, which can apply to many types of abnormal locomotion, from mild to severe. The stated criteria for “jerky-unc” are that “animals exhibit a lack of smooth locomotion with frequent abrupt stops and start before and after a touch on the head.” This confuses me – if the stops and starts occur both before and after a touch on the head, and the worm fails to reverse direction, this implies that the animals are touch-insensitive. In Figure 3C, the terminology “*C. elegans* motor disruption” is used. In some parts of the Supplemental Data spreadsheet, the term “atypical locomotion” is used. I think that either one of these descriptors would be preferable to “jerky-unc.”

Our Response: For clarification, the Nemacol-exposed animals that were originally described as ‘jerky-Unc’ are not touch insensitive. We agree with Dr. Rand that it is a subjective descriptor and that it is a challenge to come up with an appropriate and distinctive terminology to concisely describe this phenotype. Given that the recommended alternatives “*C. elegans* motor disruption” and “atypical locomotion” can apply to any uncoordinated movement, we have elected to refer to the phenotype as pausing uncoordinated movement or ‘Punc’, which also has the advantage of being succinct. We describe this in the opening results paragraph as follows, ‘Within minutes of exposure, Nemacol-1 causes frequent abrupt pausing during locomotion, a phenotype that we will refer to here as pausing uncoordinated (Punc) locomotion’ (Fig. 1a-d; Supplemental Movies 1-3).’ We have changed the ‘jerky-Unc’ to ‘Punc’ throughout. Note that the reader can now inspect the Punc phenotype with the new movies we have added (Supplemental Movies 2 and 3). We hope this is acceptable to Dr. Rand.

Reviewer #2- Comment 2: Paralyzed: One might predict that decreasing the release of an excitatory neuromuscular transmitter would lead to a flaccid paralysis, but this is not the case. Severe (but non-null) *unc-17* and *cha-1* mutants are not flaccid at all, but rather are quite uncoordinated, often tightly coiled (such as the upper worm in the video), and jerky (almost ratchet-like) when backing. Unfortunately, in the manuscript the terms paralysis and flaccid paralysis seem to be used interchangeably. (For a good example of a truly flaccid phenotype, look at *unc-52* homozygotes.) If in fact, Nematicol exposure produced a really flaccid phenotype, it would be unexpected and of great interest.

Our Response: Nematicol-1-treated worms that are paralyzed are not flaccid like *unc-52* mutants. They also not rigid either. They are similar to wild type in their rigidity but fail to move on their own during the observation period and fail to move when prodded on the head. We have therefore removed the term flaccid in the text and simply refer to the phenotype as paralysis. To help clarify, we define the paralysis in the revised first paragraph of the results section as follows (the relevant section is underlined) and have added two new Supplemental Movies (6 and 7):

Over the course of four hours, the Punc phenotype transits to tight coiling and paralysis (whereby a fraction of the animals fail to locomote during the observation period, even when prodded on the head; see Supplemental Movies 6 and 7), and then gradually transits back to the Punc locomotion.

Reviewer #2- Comment 3: Coiled: This is an accepted descriptor, and is straightforward to score. However, the stated criteria for “paralysis” (“animals...fail to reverse upon light touch of the head with a platinum wire”) would also apply to tightly coiled animals. It looks like every worm was assigned to one of these 3 phenotypes or wild-type. What about animals that didn’t fit into any of the 4 phenotypes (lethargic animals, “loopy” Uncs, other types of non-jerky Uncs)?

Our Response: First, the phenotypic bins are mutually exclusive. We clarified this in the revised ‘Scoring of Motor Phenotypes’ methods section by adding ‘either’ in the relevant sentence (i.e. ‘Animals were scored as having either Punc locomotion if...’. We also provided additional clarity by adding the following sentence to the same methods section: ‘Coiled animals were not scored as paralyzed despite sometimes not responding to light touch.’

Second, we did not encounter animals that deviated noticeably from the four bins in our analysis (wt, Punc, coiled or paralyzed).

Reviewer #2- Comment 4: Also, do the 3 non-wild-type phenotypes represent a gradient of severity?

Our Response: Our subjective opinion is that the paralysis is more debilitating than the coiled phenotype, which is more debilitating than the Punc phenotype. However, we do/did not feel compelled to make such an argument in the manuscript.

Perhaps we are incorrect in assuming that this question may be prompted by our focus on Nematicol-1 when perhaps Nematicol-2 induces stronger phenotype. However, our previous analysis indicated that Nematicol-2 (aka ‘wact-6’ in our large scale screen presented in Burns et al., 2015) kills human HEK293 cells, so we were discouraged from pursuing Nematicol-2 in any detail.

Reviewer #2- Comment 5a: Phenotypic measurements, statistics, and plotting: The color density plots in Figures 2B, 2C, 4A, and S2A are confusing.

Our Response: We are confident that the confusion comes from the fact that the scale bars for the original heatmap charts in Figures 2b, 2c, and S2 were inverted. We deeply regret this error, apologize for the confusion, and have fixed the error in the revision.

In addition to fixing the scale bar, we have added the mean values within each cell of the heatmap to clearly illustrate the impact of the conditions on the phenotype being measured.

We note that the use of double axes heatmap plots are popular in many fields, including genomics and chemical biology. It is a spatially economical way of presenting a lot of numerical information. A couple of recent examples include:

Cheng et al., 2019 Nature Communications (<https://www.nature.com/articles/s41467-019-09186-x>)

Zost et al. 2020 Nature (<https://www.nature.com/articles/s41586-020-2548-6>)

Reviewer #2- Comment 5b: The authors perform statistical analysis on a large set of what are admittedly subjective descriptions, and plot the results in a confusing (to me, at least) way. This isn't necessary, because there's no need to provide detailed quantitative measurements of the effects of these drugs on behavior. For example, the first set of results demonstrates the *unc-17* mutants are hypersensitive to Nemaicol; the second set of results demonstrates that *md414* mutants are partially resistant to Nemaicol; and the third set of results demonstrates that Nemaicol-1 can suppress the paralysis induced by AChE inhibitors. In each case, quantitative details (*e.g.*, precise EC50 values) are less important than the (qualitative) conclusion. For qualitative conclusions, comparative images of worms are better than graphs – readers can grasp the conclusions more easily and in less time. See, for example, Figures 4A, 5, and 6 from the Sluder paper.

Our Response: First, we have included pictures of the phenotypes that reflect several of the relevant genetic interactions with nemaicol in the revised Figure 2, including induction of coiling in wt, slight enhancement of coiling in *e245* (which also serves as a positive control for coiling), *md414* resistance of Nemaicol, and most dramatically, suppression of the AChE inhibitor trichlorfon.

Second, we have included EC50s and more statistical analyses throughout the revised figures and paper. EC50s and stats are standard forms of analyses when dealing with the effects of compounds and it is something that readers in the field of chemical genetics and early stage drug discovery will want to see.

Third, we have added six new Supplemental Movies that show the phenotypes induced by Nemaicol-1. We hope that these improvements are satisfactory.

Reviewer #2- Comment 5: Lines 63-65: “The neurotransmitter acetylcholine (ACh) is the primary signalling molecule in **most** animals that triggers muscle contraction.” Not in all animals – for example, insects use glutamate as the excitatory neuromuscular transmitter.

Our Response: We thank Dr. Rand for correcting this point and have made the suggested change.

Reviewer #2- Comment 6: Line 75: "...is inhibited by vesamicol and by the relatively novel spiroindoline scaffold"

Our Response: We have made the suggested change.

Reviewer #2- Comment 7: Lines 226-229: "VACHT mutant residues that confer resistance to vesamicol or the spiroindolines disrupt the ability of these molecules to interact with VACHT, but reduce acetylcholine interaction and lead to obvious locomotory defects in *C. elegans*."

This statement is incorrect. According to the Sluder paper, the spiroindoline-resistance mutants they obtained "had no obvious phenotype as homozygotes."

Our Response: We have changed this sentence to the following: 'The VACHT mutant residue that confers resistance to vesamicol disrupts the transporter's ability to interact with the inhibitor, reduces the transporter's interaction with acetylcholine and lead to motor defects in *C. elegans*.' The accuracy of each of the clauses in this revised sentence is described below:

First, 'The VACHT mutant residue that confers resistance to vesamicol disrupts the transporter's ability to interact with the inhibitor...' is supported by the 2001 Zhu et al paper wherein it reports that the C391Y mutation in the human VACHT, which corresponds to the worm *unc-17(md414)* C370Y mutation abolishes the transporters affinity for vesamicol. They write, '*the C391Y mutant did not bind vesamicol (Table I), and acetylcholine transport in membrane preparations from this mutant was not inhibited by vesamicol (Fig. 6).*'

Second, the clause '... reduces the transporter's interaction with acetylcholine...' is supported by the Zhu et al 2001 paper: '*The Km of VACHT for acetylcholine was increased in the four examined mutants G233F (2 mM), C391Y (3 mM), A228V (6 mM), and S252F (8 mM) relative to wild-type (1 mM).*' Furthermore, the *unc-17(md414)* C370Y mutation was picked up in screens for mutants that resist the acetylcholinesterase inhibitor, indicating that less acetylcholine is released from the synapse in the *md414* mutants.

Third, '...and lead to motor defects in *C. elegans*.' Here, the Zhu et al., 2001 paper reports that *unc-17(md414)* has, '*...moderately impaired neuromuscular function.*' This point is illustrated in Figure 3 of that paper, which shows obviously less pharynx pumping and thrashing. The Sluder study did not report any careful analysis of motor activity and the relevant statement is likely a reflection of antidotal observations of worm behaviour on plates. We hope our changes to this sentence are acceptable.

Reviewer #2- Comment 8: Lines 229-231: "Missense mutations that alter the ability of VACHT inhibitors to interact with the transporter will likely confer a clear selective disadvantage to worms." This line of reasoning is flawed. Although animals carrying such VACHT/UNC-17 mutations may be at a selective disadvantage in the absence of the drug, in the presence of the drug they are quite likely to outcompete wild-type animals. This is why pesticide-resistance is such a problem in the field.

Our Response: We agree with Dr. Rand's assessment of the logic under constant pressure of an anthelmintic/nematicide. However, modern anthelmintic practices are turning to the use of maintaining a 'wild type' or 'drug-sensitive' reservoir within infected populations such that drug-resistant allele

frequencies do not take over the parasite population. During periods where animals are prescribed to be off of the compound, the drug-sensitive allele can prevail. In this way, parasite infestation is appropriately managed (but not eliminated, which is nearly impossible).

In this light, we have changed the relevant sentence (and added another) as follows:

‘Hence, missense mutations that alter the ability of VAcHT inhibitors to interact with the transporter will likely confer a clear selective disadvantage to worms in the absence of NemaCol pressure. Ensuring persistence of drug-sensitive alleles within population is becoming a more common practice of parasite management⁵⁹.’

Reviewer #2- Comment 9: Lines 284-403: This material is outside my field of expertise, and I am not able to comment on it.

Our Response: We are confident in the chemistry methods as they were supervised by one of the preeminent organic chemists Prof. Mark Lautens and his student Rachel Baker who is now at CalTech.

Reviewer #2- Comment 10: Line 82: “...which is a phenotype shared by mutants of the CHA-1 **choline** acetyltransferase enzyme...”

Our Response: We thank Dr. Rand for correcting this point and have made the change.

Reviewer #2- Comment 11: Line 683: Change “Oklahoma (Center for Neuroscience)” to “(Oklahoma Center for Neuroscience)”

Our Response: Are sure you don’t want to be remembered as James Rand Oklahoma? (Sorry for the typo! It is fixed.)

Reviewer #2- Comment 12: Reference #31 is a duplicate of reference #27. Reference #36 is a duplicate of #32. There may be other examples as well. Reference #60 provides 4 authors and a publication year, but no title and no citation.

Our Response: We have reviewed the references in detail and have corrected them in the revised version.

Reviewer #2- Comment 14: Figure 2A: The data cluster at 0 μ M NemaCol is missing the data for *e327*, *e795*, and *md414*, all of which are significantly uncoordinated.

Our Response: Upon reinspection (the student who did the original analysis (Pyche) has since graduated and is now in medical school), we find that *md414* has no obvious uncoordination on solid substrate with *E. coli*. We do see that *e327* mutants are uncoordinated; they are a little slow and have slightly exaggerated bends, but neither *md414* nor *e327* exhibit the Punc, coiled or paralysis phenotypes.

These observations are consistent with comments made in the 2001 Zhu et al JBC paper; ‘*The “plate clearing” assay is affected by both the generation time and the fecundity of the mutants. By each of*

these criteria, *e327* (CeS211F), *e795* (CeA206V), and *p1160* (CeC230F) form a graded series of increasing severity, whereas *md414* (CeC370Y) homozygotes have approximately normal growth and only moderately impaired neuromuscular function.'

To clarify these points regarding *md414* and *e327* on Figure 2a, we have added 'np' to the graph and added the following note in the legend: 'Np; the animals fail to exhibit any of the Punc, paralyzed or coiled phenotypes.'

By contrast, when re-examining *e795*, we do see obvious Uncoordination with large body bends and a few of the animals were coiled upon prolonged observation. This prompted us to redo the analysis of *e795* along side wild type controls. We have updated Figure 2a accordingly. None of the conclusions, however, have changed- *e795* remains significantly hypersensitive to Nematicol-1 relative to the wild type control ($p < 0.001$)- see Figure 2a.

Reviewer #2- Comment 15: Figures 2B, 2C, S2A: the horizontal color-density scale is backwards.

Our Response: We apologize for the errors and have corrected the oversights in the revised manuscript.

Reviewer #2- Comment 17: Video: The tightly coiled animal appears quite similar to severe loss-of-function *unc-17* or *cha-1* mutants. The other two worms do not display normal locomotion, but I'm not sure I would describe their movement as jerky. In any event, they do not appear to be seriously incapacitated.

Our Response: We agree with Dr. Rand's assessment. To add clarity, we have made comments in the relevant Supplementary Movie of the coiling animal (now called Supplementary Movie 4) description indicating which is phenotypically coiled and which is not. We have also added several additional Supplementary Movies of wild type animals, including one of animals on DMSO (drug solvent) only, two of animals on Nematicol-1 exhibiting the Punc phenotype, and two of animals on Nematicol-1 exhibiting paralysis and being poked.

Reviewer #2- Comment 18: Lines 225-231 (the 2nd, 3rd, and 4th sentences of the final Discussion paragraph) are factually incorrect, and should be eliminated.

Our Response: This is the same comment as Dr. Rand's comment #7 above, which we hope we have satisfactorily addressed.

Reviewer #2- Comment 19: Figures 2B, 2C, 4A, and S2A should be eliminated, and replaced by images of worms at specific data points. See, for example, Figures 4A, 5, and 6 from the Sluder paper. **HOWEVER**, if this recommendation is unacceptable for some reason, my (reluctant) fallback suggestion for these figures is to make them at least less confusing by following the example used in Figure S3A, and entering a specific value in each cell.

Our Response: As discussed in our responses to comments 5a and 5b (above), we have happily included new photos and new movies of the phenotypes. We have also elected to keep the detailed analyses in the figure panels referred to above, but have added additional annotation to help make the figures easier to absorb. We think that these suggestions improve clarity and thank Dr. Rand for the suggestions.

Reviewer #3 (Remarks to the Author)

In the present study, the authors investigated the anthelmintic activities of Nemacol, a nitrophenyl-piperazine scaffold. Starting from 4 compounds selected from a screen performed in another study, they use *C. elegans* as a model to decipher the mode of action of these drugs, highlighting the vesicular acetylcholine transporter UNC-17 as the main target. Nemacol-1 was investigated in more details, including the role of drug metabolizing CYP450 in its transient activity on the worms. Subsequently the authors investigated Nemacol analogs for acute motor tests in *C. elegans*, *P. pacificus* and the Dog heartworm *D. immitis*. Finally, the authors tested combinations of Nemacol with the widely used anthelmintic Ivermectin and reported a promising synergic effect.

This paper is of great interest as it reveals a novel class of anthelmintic for which there is an urgent need. Even though the present content lays a strong basis for an attractive and original article, I would recommend important modifications to the manuscript.

Our Response: We thank the reviewer for taking the time to review our work. We hope they recognize the amount of effort (experimental and otherwise) that we have gone through to address their comments.

Reviewer 3- Comment 1: First: As mentioned by the authors (line 94) the four molecules which paved the way for this study have been identified from a previous screen (ref 10). In the mentioned paper, the authors took care of investigating the activity of the compounds on a range of parasitic nematode species as well as off-target species, thus providing critical information on both the spectrum of activity and the potential toxicity for vertebrates.

In the present work, the authors did not refer to the molecule's nomenclature used in this "reference" paper. Therefore, in order to support their claim concerning the potential use of Nemacol as an anthelmintic treatment, the authors should provide for each compound the information concerning both the activity observed on the other parasitic nematode species as well as the data concerning the off-target species.

Our Response: We refer to the 2015 Burns *et al* screen in the introduction and results sections in the original and revised manuscript, but Reviewer 3 is correct- we failed to disambiguate which molecules from that screen are the nemacols. All molecules in our worm active (wactive) library that is described in Burns *et al.*, 2015 have a 'wact' prefix followed by a unique number. We have chosen to give small molecule families from that library that we are characterizing in detail more unique names to distinguish them from other structural families.

We have corrected this oversight by modifying the second sentence of the first paragraph of the results section to the following:

'All four molecules, which we previously referred to as wact-45, wact-6, wact-46, and wact-47, share a 1-ethyl-4-(4-nitrophenyl)piperazine substructure. We have renamed these molecules Nemacol-1, -2, -3, and -4, respectively.' We hope this is satisfactory.

Reviewer 3- Comment 2: Note that the authors mentioned in the discussion part (line 191) that “assays performed against multiple parasitic nematode species over an equivalent timespan indicate that NemaCol’s effects can perdure in parasites”, however they did not mention a corresponding reference and in the present study there is no data supporting this claim.

Our Response: The sentence that Reviewer 3 cites here is referring to the data we present in this current (2022/2023) manuscript (i.e., original and revised Figure 3c, and the newly added Figures 4d and S6). In other words, the data of how the parasitic nematodes that we have assayed here (*Dirofilaria* and *Haemonchus contortus*) support the claim that NemaCol’s effects can perdure in parasites.

To add clarity to this point, we have modified the relevant sentence to the following (additions are underlined): ‘However, assays performed here against multiple parasitic nematodes over an equivalent timespan (or longer) (see Figs. 3c, 4d, and S6) indicate that NemaCol’s effects can persist in parasites.’

Reviewer 3- Comment 3: Second: The work on *Dirofilaria immitis* is incomplete. Based on the main objective of the paper, there are too few attempts to transpose the results from *C. elegans* to the parasite species. For *C. elegans*, the authors described in details the different phenotypes (over time) associated with the drug exposure. Here the same information should be provided for *D. immitis*.

Our Response: We have addressed this point experimentally in three ways.

First, we asked whether we could recapitulate NemaCol activity against *Dirofilaria* in another laboratory. Of note, our first *Dirofilaria* analyses was in collaboration with Daniel Kulke while he was at Bayer; he has since moved on and we no longer have access to those assays. To address the Reviewer 3’s comments, we therefore initiated a new collaboration with Mostafa Zamanian’s lab at the University of Wisconsin. As can be seen from Supplemental Figure 6, we were able to recapitulate NemaCol-1’s activity against *Dirofilaria* in a detailed dose-response analysis, albeit the potency is slightly lower in the second lab.

Second (and prompted by Reviewer 3’s additional comments below), we tested whether we could enhance ivermectin’s (IVM) efficacy against *Dirofilaria* in an *in vitro* assay. We did this despite knowing in advance that IVM does not behave as expected against *Dirofilaria in vitro* (see Bourguinat et al., 2011; Evans et al., 2013; Storey et al., 2014¹). *In vitro*, IVM is only effective at concentrations >1000 fold higher than what is effective *in vivo* (Maclean et al., 2017) and what we see with IVM treatment with *C. elegans in vitro*, prompting some to speculate that IVM is hitting non-canonical targets in the *in vitro* assay (Maclean et al., 2017). Regardless, we thought it was worth an attempt. As illustrated below, we observed no impact of an EC30 of NemaCol-1 on IVM’s ability to paralyze *Dirofilaria* at either the 24 or 72 hour timepoint. This result is expected if IVM *in vitro* effects on *Dirofilaria* at these high

¹Bourguinat, C., Keller, K., Blagburn, B., Schenker, R., Geary, T.G., and Prichard, R.K., 2011.

Correlation between loss of efficacy of macrocyclic lactone heartworm anthelmintics and P-glycoprotein genotype, *Veterinary Parasitology*, 176 (4), 374-381.

Evans, C. C., Moorhead, A.R., Storey, B.E., Wolstenholme, A. J., and Kaplan, R. M., 2013. Development of an *in vitro* bioassay for measuring susceptibility to macrocyclic lactone anthelmintics in *Dirofilaria immitis*. *Int. J.Parasitology: Drugs and Drug Resistance*, 3, 102-108.

Storey, B., Marcellino, C., Miller, M., Maclean, M., Mostafa, E., Howell, S., Sakanari, J., Wolstenholme, A., and Kaplan, R.Utilization of computer processed high definition video imaging for measuring motility of microscopic nematode stages on a quantitative scale: “The Worminator”. *Int. J.Parasitology: Drugs and Drug Resistance*, 4(3), 233-243.

concentrations are not exerted through canonical targets. We elected to not include this data in the manuscript because of IVM's confounding effects against *Dirofilaria in vitro*.

Third, in anticipation that our analysis of drug interactions with *Dirofilaria* would not yield meaningful results, we reached out to Alexandre Vernudachi at Invenesis Inc, who has helped develop sensitive assays for the impact of drugs on the movement of the ruminant nematode parasite *Haemonchus contortus*. He first tested a few of our Nemacol analogs to determine which might be active against *H. contortus*. We knew from our Burns *et al* 2015 work that Nemacol-1 (aka wact-45) did not have good *in vitro* activity against *H. contortus*, so we sent him a few additional analogs, one of which worked well (nemacol-53) (new Figure 4d). We then tested the EC30 of Nemacol-53 for its ability to enhance IVM's efficacy against *H. contortus* and indeed observed significant enhancement ($p < 0.002$) (new Figure 4e). Consequently, we have added the following as the last paragraph in the results section:

Finally, we wanted to test whether Nemacol can enhance the effects of ivermectin in the context of a parasitic nematode. Because *Dirofilaria immitis* is known to be refractory to ivermectin in *in vitro* assays (PMIDs: 29143656, 21300438, 24533299), we asked whether enhancement may be seen with the ruminant nematode parasite *Haemonchus contortus* (PMID 23998513). A small survey indicated that Nemacol-53 has activity against *H. contortus*, with an EC30 of 26 μM (Fig. 4d). This concentration of Nemacol-53 is able to significantly enhance ivermectin's ability to paralyze *H.*

contortus in vitro ($p < 0.002$) (Fig. 4e). These data indicate that NemaCol may have additional utility in its ability to sensitize nematodes to one of the most widely used anthelmintics in the world.

Reviewer 3- Comment 4: The description of the screening approach on *μ*filaria is minimalistic, there is no information about the precise parameters that have been monitored using the imaging approach.

Our Response: We apologize for this. We have revised this section to include more details, including an additional sentence to describe the imager and a reference that describes the imager and data capture in even more detail. We hope this is satisfactory. For your convenience, we have pasted that section here:

***Dirofilaria immitis* Culture and Small-Molecule Assays**

We conducted experiments with the Missouri isolate of *Dirofilaria immitis* in two labs. In the first set of experiments, shown as part of Figure 3c, *D. immitis* microfilariae and larval stage 3 (L3) worms were assayed in the laboratories of Bayer Animal Health GmbH (Monheim, Germany) in accordance with the local Animal Care and Use Committee and governmental authorities (LANUV#200/A176 and #200/A154). For microfilariae immobility assays, approximately 250 freshly purified microfilariae were cultured in single wells of a 96-well microtiter plate containing supplemented RPMI 1640 medium (PMID 28717951, 34339934). Compounds were added in the following concentrations: 50 μ M, 10 μ M, 2 μ M, 0.4 μ M, 0.08 μ M, 0.016 μ M and 0.0032 μ M. Microfilariae exposed to medium substituted with 1% DMSO were used as negative controls. Motility of microfilariae was evaluated after 72 hours of drug exposure using an image-based approach – Dirolmager, developed by Bayer Technology Services. As described in detail in (PMID 28717951, 34339934), the Dirolmager is an automated high-throughput platform that allows for high-resolution optical imaging of an entire 96-well microtiter plate. Data are reported as the EC50 (μ M) calculated from the tested concentration series.

For the *Dirofilaria immitis* larval development assays, freshly isolated L3s were cultured in wells of a 96-well microtiter plate with 10 L3s per well. All wells contained supplemented RPMI 1640 medium (PMID 28717951, 34339934) and a test compound at one of the following concentrations: 10 μ M, 2 μ M, 0.4 μ M, 0.08 μ M, 0.016 μ M and 0.0032 μ M. L3s exposed to DMSO only (1%) were used as negative controls. All drug concentrations were tested in duplicate and drug effects were evaluated after 72 hours of incubation, after which DMSO-only controls are L4s and the shed L3 cuticles are evident in the wells. Compounds that have dramatic acute effects retard or arrest the growth of the nematodes and they remain L3s, while other compounds have less severe effects and the nematodes grow to the L4 stage. Motility was scored for each sample in the following standard manner: The 20 worms total from the two duplicate sample wells are considered. If only one of the 20 worms moves (as detected by pixel displacement with the Dirolmager), the score for that sample is considered as 5% motility. Data on a given day were considered as valid only if the DMSO-only controls exhibited at least 90% motility. Data is then reported as the EC50 (μ M) calculated from the tested concentration series.

We also conducted *Dirofilaria immitis* experiments in the lab of M. Zamanian (University of Wisconsin). For this experiment, shown in Figure S6, Microfilaremic blood was obtained from the NIH/NIAID Filariasis Research Reagent Resource Center (FR3) (PMID: 22140585). Blood was drawn and shipped approximately 24 hours before use. Upon arrival, blood was warmed to 37°C then combined with a 0.85% sodium chloride and 0.2% saponin solution in a 1:11 volumetric ratio and incubated in a 37°C water bath for 15 minutes. The hemolyzed solution was passed through a 25mm 5.0 μ m pore size syringe filter. The used filter disks were transferred to petri dishes filled with RPMI 1640 culture media with 1% penicillin/streptomycin (0.1 mg/mL) and incubated at 37°C for 2 hours while microfilariae (mf) separated from the disks. Disks were discarded and mf titered to 5 microfilariae/ μ L.

Aliquots of titered mf were incubated on a heating block at 60°C for 1 hour to produce heat killed positive controls. One μ L of 100x drug was aliquoted to each well of a 96-well plate; 100 μ L of live mf were added to treatment wells and 100 μ L of heat killed mf were added to positive control wells (for a total of 500 mf/well). Plates were sealed with a breathable plate cover and maintained at 37°C and 5%

atmospheric CO₂. Videos of the plates were taken every 24 hours for 72 hours using an ImageXpress Nano (4x, 10 frames per well). A full protocol for the imaging process can be found here: <https://doi.org/10.21203/rs.3.pex-1916/v2>. After acquiring the 72-hour timepoint video, viability staining was performed using the CellTox Green kit (Promega); a full protocol for this procedure can also be found at the previous link. Image analysis and the subsequent measurement of optical flow and fluorescence was performed using wrmXpress v1.3.02 (PMID: 36399491).

Reviewer 3- Comment 5: Apart from the EC₅₀ values (without SE information...) there is no data nor statistical analysis provided. In contrast with the *C. elegans* experiments, the authors did not clearly indicate the number of independent trials (independent batches of worms) used for *D. immitis* experiments.

Our Response:

Revised Figures 1a-e (and/or legend) reports means, EC₅₀ values, 95% confidence intervals, statistical analyses, and N and n values.

Revised Figure 2a (and/or legend) reports statistical analyses, means, standard error of the mean, and N and n values.

Revised Figures 2c and 2d (and/or legend) reports means for each drug combination, EC₅₀s for the curves along with 95% confidence intervals, statistical analyses of the EC₅₀s, and N and n values.

Revised Figure 2e (and/or legend) reports means and standard error of the mean for individual data points, EC₅₀s for the curves along with 95% confidence intervals, statistical analyses, and N and n values.

Revised Figure 3a (and/or legend) reports means and standard error of the mean for individual data points, EC₅₀s for the curves along with 95% confidence intervals, statistical analyses, and N and n values.

For Figure 3b, the means and N and n values are presented in Figure 3c, and the 95% confidence intervals are presented in Supplementary Table 1.

Revised Figure 3c reports means and 95% confidence intervals for the *C. elegans* and *P. pacificus* assays. The legend reports the N and n values for the *Dirofilaria* survey. (Note that greater N values are used for the detailed dose-response assay shown in Supplemental Figure 6). The confidence intervals for the rat VACHT Ki values are presented in Supplemental Table 1.

Revised Figure 4a (and/or legend) reports means for each drug combination, EC₅₀s for the curves along with 95% confidence intervals, statistical analyses of the EC₅₀s, and N and n values.

Revised Figures 4c-e reports means and standard error of the mean for individual data points, EC₅₀s for the curves along with 95% confidence intervals, statistical analyses, and N and n values in the legend.

Similar analyses are reported for the data presented in the Supplemental Figures. We hope this is satisfactory.

Reviewer 3- Comment 6: Concerning the L3 (for which surprisingly the results have been reported as L4 in fig.4C), the authors mentioned a developmental assay whereas they performed a motility assay. The sentence (lines 432-433) should be reformulated as we could only hardly guess what the authors meant while describing the test.

Our Response: First, we apologize for the confusion on the details of the *Dirofilaria* larval assay. As described in the more detailed method section on *Dirofilaria*, the assay starts with L3s, but by 72 hours later, solvent only controls are L4s. Compounds that have dramatic acute effects retard or arrest the growth of the nematodes and they remain L3s, while other compounds have less severe effects and the nematodes grow to the L4 stage. We have since changed how we call the assay in the Figure as ‘*Dirofilaria immitis* L3/L4’.

Second, as described in the methods section, while the assay takes place over a developmental time frame, the data that is actually being collected is the movement of the parasites.

Third, we apologize for the confusion in the *Dirofilaria* methods section- this is a result of English translation errors. We have since clarified the methods in the revision (see above) and hope that it is now clear.

Reviewer 3- Comment 7: On *D. immitis*, the scoring was performed after 72H exposure whereas for *C. elegans*, three among the four tested drugs (in the first part of the manuscript), either lost or had a reduced activity after 4h. This should be explained.

Our Response: This comment is related to Reviewer #3’s comment #2 above. In short, the nematodes studying in detail in *C. elegans* are metabolized over time in *C. elegans*, but evidently not to such an extent in the *Dirofilaria* and *Haemonchus* parasites so as to abolish their activity. Hence the revised sentence in the discussion, ‘However, assays performed here against multiple parasitic nematodes over an equivalent timespan (or longer) (see Figures 3c, 4d, and S6) indicate that Nematicol’s effects can persist in parasites.’

Reviewer 3- Comment 8: Finally, whereas the search for synergy identified a positive interaction between Nematicol-1 and IVM in *C. elegans*, could the authors justify why there was no attempt to transpose this important result to *D. immitis*, knowing that Macrocytic lactones are used for the treatment of this parasite. If such “synergy” could be confirmed in a parasitic nematode species, this would represent an important proof of concept thus increasing the impact of the paper.

Our Response: We have addressed this point in our response to this reviewer (Reviewer #3)’s comment #3 above.

Reviewer 3- Comment 9: Third: search for synergy. Whereas the discovery of drug combinations leading to a lethal phenotype is of major interest, this part needs to be improved to further support the authors claim. The authors should justify why they use a developmental assay (72h) instead of the motility assay. Indeed, this could be misleading as the authors decided to use Nematicol-1 for which they demonstrated a transient activity (24h) and then provide an evidence that the same drug is able to kill some worms at 60µM after 72h...this observation should be mentioned in the first part of the manuscript.

Our Response:

The Reviewer’s comment on synergy is addressed in our response to their comment #3 above.

The Reviewer's comment on the issue of a 72 hour development assay versus a motility assay is addressed in our response to their comment #6 above (in short, the assay is in fact a motility assay).

Reviewer 3- Comment 10: Line 170-172 the authors state that "combination of IVM and Nematicol could yield effective killing of *C. elegans* at concentrations that had little effect on their own". A "little effect" could be interpreted in many different ways. Unfortunately, Fig4A is very difficult to read and it is impossible to accurately attribute a value to the different purple nuances and distinguish potential additive effects, synergies or even absence of effect. The authors should provide (at least for a set of relevant combinations) a detailed statistical analysis to support their claim.

Our Response: We address this comment in multiple ways.

First, we have added the mean values of the percentage viability in each cell that represents the various drug combinations in Figure 4a (and S2a and S7a).

Second (and related to the reviewer's comment #5 above), we have added EC50s and 95% confidence intervals along with statistical analyses to the Figure 4a.

Third, we have revised the relevant section to read as follows (changes are underlined):

Indeed, we found that in three-day liquid viability assays (see methods), combinations of ivermectin and Nematicol could yield effective killing of *C. elegans* at concentrations that had negligible effect on their own (for one example, see 30 μ M Nematicol-1 and 15 nM ivermectin in Fig. 4a). Nematicol-1 significantly lowered ivermectin's EC50 by 3 fold ($p < 0.001$; Fig. 4a). Furthermore, the combination of the two compounds yields a global Zero Interaction Potency (ZIP) synergy score of 20.0, which is beyond the ZIP score threshold of synergy (10^{42-44} (see the intense area of Fig. 4b).

We hope these changes are satisfactory.

Reviewer 3- Comment 11: Note that Fig4C without highlighting the zone of interest doesn't bring a lot of help for the reader.

Our Response: We have moved Figure 4c to Figure 4b. We have clarified the zone of interested to the reader by writing (addition is underlined): 'Furthermore, the combination of the two compounds yields a global Zero Interaction Potency (ZIP) synergy score of 20.0, which is beyond the ZIP score threshold of synergy (10^{42-44} (the intense red area of Fig. 4b).

Reviewer 3- Comment 12: The Fig4 B is much more convincing (however note that based on the dose-response curve, the EC50 values for IVM should be in nM and not in μ M) and lays a strong basis for the demonstration. The authors should provide the dose response curve for Nematicol-1 with fixed concentrations of IVM.

Our Response: We apologize for the error in noting the units of concentration and have corrected it in the revised version (note that original Figure 4b is now Figure 4c).

We make no explicit claims about IVM's ability to enhance nematicol's activity (beyond noting the synergy) in the original or revised manuscript. We therefore elected not include graphs for the dose response curve of nematicol-1 with fixed concentrations of IVM. However, we have added the mean

values for the individual data points of the worms' response to the compounds in Figure 4a for clarity (and from which curves could be derived for those who are curious). We hope this is satisfactory.

Reviewer 3- Comment 13: ...and it would be also of major interest to investigate the effect of Nemacol-1 on an IVM resistant *C. elegans* strain such as DA1316 mutant in order to mirror the approach presented in Fig4B.

Our Response: At the reviewer's suggestion, we have performed this experiment using the same concentrations used in Figure 4 (see Supplementary Figure 7). We report the results in the third last paragraph of the result section as follows:

Previous work has identified a number of targets for ivermectin in *C. elegans*, including the glutamate-gated chloride channels AVR-14, AVR-15, and GLC-1⁴³. We investigated whether the synergy observed between Nemacol-1 and ivermectin in wild type animals is maintained in the *avr-14; avr-15; glc-1* triple *C. elegans* mutant. As expected, the triple mutant lost responsiveness to ivermectin at the concentrations tested (Fig. S7a). For example, Fig. S7b compares ivermectin's effects on the triple mutant (without Nemacol-1; Fig S7a) to ivermectin's effects on the wild type (without Nemacol-1; Fig 4a) ($p < 5.5 \times 10^{-11}$). Notably, the double drug treatment remains synergistic in the triple mutant (Fig. S7c), albeit 0.6-fold less compared to how the drugs behave in the wild type (compare with Fig. 4b). This suggests that some of the observed synergy between Nemacol-1 and ivermectin derives from ivermectin's interaction with other targets beyond AVR-14, AVR-15, and GLC-1. This insight is consistent with previous findings showing that additional components, including GLC-3 and perhaps UNC-7, also mediate ivermectin sensitivity in *C. elegans*⁴³⁻⁴⁵.

Reviewer 3- Comment 14: The authors should justify their choice concerning the focus they made on Nemacol-1, which harbours a transient activity, instead of Nemacol-2 for which there was no apparent drastic reduction of activity after 4h while sharing similar EC50 values with Nemacol-1 on *C. elegans*.

Our Response: To address this comment, we added the following new paragraph after the first in the results section:

Of the four original Nemacol hits, analogs 1 and 2 induced the strongest phenotypes in *C. elegans*. However, Nemacol-2 (aka wact-6) demonstrated adverse effects in our previous counter-screens (Burns *et al.*, 2015). We therefore focused on nemacol-1 for much of the analyses presented below.

Reviewer 3- Comment 15: Note that a test at 1440 min (as for Nemacol-1) would be very informative for Nemacol-2.

Our Response: We have addressed this comment by performing 24 hour analyses for all of the compounds presented in the revised Figure 1. Please see the figure for details.

Reviewer 3- Comment 16: Similarly, the authors should justify the selection they made for the drugs used in the acute tests performed on *D. immitis*.

Our Response: The *C. elegans*, *Pristionchus* and *Diriofilaria* assays reported in the relevant Figure 3c were done with molecules that were available to us at the time. In a few cases, we ran out of the molecule at the time the assays were being performed. The rat VACHT Ki assays are very expensive and time-consuming, so we selected molecules that represented diverse structures that had activity in the other assays. We justify this in the respective legend by writing the following:

‘The grey cells lack data because of either insufficient material at the time the assays were performed, lethality in non-targeted systems, or because of the expense associated with the rat VACHT binding assays.’

Reviewer 3- Comment 17: What was the reason for not investigating Nemacol-2 on the parasite whereas it appeared to more relevant than Nemacol-1 for the control of a parasite species that cannot be expelled from the host?

Our Response: We have addressed this comment in response to this reviewer’s comment #14 above.

Reviewer 3- Comment 18: For all the study, EC50 should be provided with their corresponding SE, indeed the variability of the results obtained with some compounds appears to be very important (Nemacol-3 and 4 for example) Dose response curves obtained during the acute tests should be provided at least as supplemental data.

Our Response: We have gone through the manuscript and have added EC50s and confidence intervals where it made sense to do so in the main figures and/or their legends (including Figures 1a-1e, 2c-2e, 3a, much of 3c, 4a, 4c, 4d, and 4e). As requested, we also added additional dose-response curves, EC50s and confidence intervals for the other experiments in Supplemental Figures 2a, 2b, 4, 5, 6, 7a, 7b and Supplemental Table 1.

Reviewer 3- Comment 19a: The discussion is currently mainly focused on the potential industrial applications (including comments on results which are not presented in the present work...L211-223).

Our Response: To address the latter point here, the only results that were not presented in the results section was the SwissADME consensus LogP and synthetic accessibility values for the nemacol analogs, which we have now added to the revised Figure 3c. We have also updated the relevant text as follows (relevant additions are underlined):

‘The high lipophilicity of the spiroindolines may have so far stifled their development into commercial products. Guidance from the European Union’s European Chemical Agency⁵¹ highlights that compounds with a Log P greater than 4 have accumulative potential in adipose tissue of animals. The primary spiroindoline lead pursued by Syngenta (SYN876 in Sluder et al., 2012²⁷) has a SwissADME predicted consensus LogP of 5.53 suggesting that this lead may have concerning accumulative potential. SwissADME is a highly cited chemoinformatics tool that is widely used in the field⁵². In contrast, 92% of Nemacol analogs had a SwissADME predicted consensus Log P \leq 4.0 with a median of 2.93⁵² (see Fig. 3c).

Relative to the structural complexity of the spiroindoline scaffold (MW 453.53 with a SwissADME synthetic accessibility score of 4.23)⁵² the Nemacol-1 structure is simpler (MW 303.4 with a synthetic accessibility score of 2.52^{52,53}) (see Fig. 3c). The higher the synthetic accessibility score, the more difficult the synthesis^{52,53}. Indeed, we have found Nemacol to have a relatively inexpensive synthesis route with

several analogs so far synthesized requiring only a two-step metal-free synthetic sequence that doesn't require purification of the intermediate (see methods).'

Reviewer 3- Comment 19b: Even though this represent an important aspect, the discussion part would largely benefit of comments and prospects about other important points raised by the present work (Nemacol detoxification, synergies, drug resistance management...)

Our Response:

With respect to Nemacol detoxification, we provide evidence that nemacol-1 is detoxified in *C. elegans* (Supplemental Figure 1), but do not investigate the phenomenon using analytical chemistry methods in other analogs or in other organisms (for reasons of cost and time). Hence, we feel that the amount of text we devote to the subject in the discussion is appropriate, where we write: 'Nemacol is detoxified in *C. elegans* over the course of 24 hours. However, assays performed here against multiple parasitic nematodes over an equivalent timespan (or longer) (see Figs. 3c and 4d) indicate that Nemacol's effects can persist in parasites.'

With respect to the management of drug resistance, we mention resistance as one rationale for the study in a couple of places, and mention resistance management only in passing in the discussion when considering the emergence of resistance to VAcHT inhibitors in the following passage: 'Hence, missense mutations that alter the ability of VAcHT inhibitors to interact with the transporter will likely confer a clear selective disadvantage to worms in the absence of nemacol pressure. Ensuring persistence of drug-sensitive alleles within population is becoming a more common practice of parasite management (Hodgkinson et al., 2019).'

Aside from these points, we feel that providing additional discussion about the management of drug resistance to be beyond the scope of our work.

With respect to synergistic interactions, we have added the following new paragraph as the last paragraph in the discussion:

In addition to having activity on its own, Nemacol enhances ivermectin activity in *C. elegans* and the ruminant parasite *H. contortus*. At the present time, the Nemacol scaffold represents only a set of tool compounds. However, its ability to synergize with one of the most effective anthelmintics increases its potential utility in the field and may potentially lower the effective dose needed of ivermectin, which is expensive to produce. In addition, Nemacol's ability to synergize with ivermectin persists even in animals that lack some of ivermectin's targets, albeit less effectively. These reasons, together with Nemacol's synthetic accessibility and the attractiveness of its target, make Nemacol an important scaffold to further develop as an anthelmintic agent.

Reviewer 3- Comment 20: Minor comments: In Fig 2 and S2 the colour code has been inversed (i.e. with dark red indicating 0% paralysis and white indicating 100% paralysis).

Our Response: We apologize for the error and have fixed the scale bars.

Reviewers' Comments:

Reviewer #1:

Remarks to the Author:

I have read the replies from the authors to my queries/questions. The authors have answered all my comments and, where appropriate, have amended the manuscript. I am now satisfied with the manuscript and require no further changes.

Reviewer #2:

Remarks to the Author:

First of all, thank you for your detailed and thoughtful "rebuttal" letter, and for the efforts you have made to improve your manuscript. I have now read your revised manuscript twice, and I particularly appreciate the images and videos of worms and drug effects - they demonstrate very clearly the effects of nemacol, and I believe that your paper will therefore be accessible to a wider audience.

When the Sluder spiroindoline paper was published, I was encouraged by the prospect of rapid advances in VChT pharmacology. I realize that the spiroindoline research is proprietary and confidential, but from my vantage point, it seems that progress has been slower than I had hoped. Your manuscript helped me to appreciate the difficulties of working with hydrophobic molecules, but it also provided a systematic approach to rational drug design, and I am once again encouraged.

Finally, a bit of cautionary advice: for some reason, when unc-17 mutants are propagated vegetatively, they are especially prone to acquire modifiers/suppressors by mutation - this seemed to happen more frequently with unc-17 than with other commonly used Unc strains. The problem is that the modifiers don't seem to suppress all the mutant phenotypes equally, so for example, some of my strains were just as slow-growing as they had been, but they moved a lot better. This was very annoying, until I started to map the modifiers, and they included some novel alleles of ace-3 and ace-2.

In any event, I strongly support publication of this manuscript.

Best wishes,

Jim Rand

Reviewer #3:

Remarks to the Author:

In the new version of their manuscript, the authors fully addressed my questions and concerns. I would like to acknowledge the authors for the impressive effort they made for generating these new sets of data. These complementary information along with the improved version of manuscript pave the way for an excellent article. Congratulations.

Response to the Reviewers Comments (verbatim):

Reviewer #1 (Remarks to the Author): I have read the replies from the authors to my queries/questions. The authors have answered all my comments and, where appropriate, have amended the manuscript. I am now satisfied with the manuscript and require no further changes.

Our Response: We thank the reviewer for their time and previous constructive comments.

Reviewer #2 (Remarks to the Author): First of all, thank you for your detailed and thoughtful "rebuttal" letter, and for the efforts you have made to improve your manuscript. I have now read your revised manuscript twice, and I particularly appreciate the images and videos of worms and drug effects - they demonstrate very clearly the effects of nemacol, and I believe that your paper will therefore be accessible to a wider audience.

Our Response: We are very happy that we have made changes that you find satisfactory. Your comments improved the manuscript greatly.

Reviewer #2 (Continued): When the Sluder spiroindoline paper was published, I was encouraged by the prospect of rapid advances in VAcHT pharmacology. I realize that the spiroindoline research is proprietary and confidential, but from my vantage point, it seems that progress has been slower than I had hoped. Your manuscript helped me to appreciate the difficulties of working with hydrophobic molecules, but it also provided a systematic approach to rational drug design, and I am once again encouraged.

Our Response: It would be lovely if our work, built on the foundations that you laid, leads to a useful VAcHT inhibitor that is used in the field.

Reviewer #2 (Continued): Finally, a bit of cautionary advice: for some reason, when unc-17 mutants are propagated vegetatively, they are especially prone to acquire modifiers/suppressors by mutation - this seemed to happen more frequently with unc-17 than with other commonly used Unc strains. The problem is that the modifiers don't seem to suppress all the mutant phenotypes equally, so for example, some of my strains were just as slow-growing as they as they had been, but they moved a lot better. This was very annoying, until I started to map the modifiers, and they included some novel alleles of ace-3 and ace-2.

Our Response: This is useful information that we will keep in mind going forward

Reviewer #2 (Continued): In any event, I strongly support publication of this manuscript. Best wishes, Jim Rand

Our Response: Thank you Jim. It has been an honour to have you critique and help us improve our manuscript.

Reviewer #3 (Remarks to the Author): In the new version of their manuscript, the authors fully addressed my questions and concerns. I would like to acknowledge the authors for the impressive effort they made for generating these new sets of data. These complementary information along with the improved version of manuscript pave the way for an excellent article. Congratulations. Cedric NEVEU (PhD).

Our Response: Thank you for the kind words Cedric and for taking the time to make our manuscript more accessible.